# Rapid O$_3$ assimilations – Part 1: background and local contributions to tropospheric O$_3$ changes in China in 2015-2020

Rui Zhu[1], Zhaojun Tang[1], Xiaokang Chen[1], Xiong Liu[2] and Zhe Jiang[1]*

[1]School of Earth and Space Sciences, University of Science and Technology of China, Hefei, Anhui, 230026, China.
[2]Center for Astrophysics | Harvard & Smithsonian, Cambridge, MA 02138, USA.

*Correspondence to: Zhe Jiang (zhejiang@ustc.edu.cn)

## Abstract

A single ozone (O$_3$) tracer mode was developed in this work to build the capability of the GEOS-Chem model for rapid O$_3$ simulation. The single O$_3$ tracer simulation demonstrates consistency with the GEOS-Chem full chemistry simulation, with dramatic reductions in computational costs of approximately 91-94%. The single O$_3$ tracer simulation was combined with surface and Ozone Monitoring Instrument (OMI) O$_3$ observations to investigate the changes in tropospheric O$_3$ over eastern (E.) China in 2015-2020. The assimilated O$_3$ concentrations demonstrate good agreement with O$_3$ observations: surface O$_3$ concentrations are 43.2, 41.8 and 42.1 ppb, and tropospheric O$_3$ columns are 37.1, 37.9 and 38.0 DU in the simulations, assimilations and observations, respectively. The assimilations indicate rapid rises in surface O$_3$ concentrations by 1.60 (spring), 1.16 (summer), 1.47 (autumn) and 0.80 (winter) ppb yr$^{-1}$ over E. China in 2015-2020, and the increasing trends are underestimated by the a priori simulations. More attention is suggested to the rapid increases in O$_3$ pollution in spring and autumn. We find stronger rises in tropospheric O$_3$ columns over highly polluted areas due to larger local contributions, for example, 0.12 DU yr$^{-1}$ (North China Plain) in contrast to -0.29 (Sichuan Basin) -0.25 DU yr$^{-1}$ (Southern China). Furthermore, our analysis demonstrated noticeable contributions of the interannual variability in background O$_3$ to the trends in surface O$_3$ (particularly in the summer) and tropospheric O$_3$ columns over E. China in 2015-2020. This work highlights the importance of rapid simulations and assimilations to extend and interpret

atmospheric $O_3$ observations.

## 1. Introduction

Tropospheric ozone ($O_3$) is produced when volatile organic compounds (VOCs) and
carbon monoxide (CO) are photochemically oxidized in the presence of nitrogen oxides ($NO_x$).
Tropospheric $O_3$ has important influences on the climate (Mickley, 2004; Iglesias-Suarez et al.,
2018), atmospheric oxidation capacity (Thompson, 1992; Prinn, 2003), human health and crop
growth (Zhang et al., 2021; Li et al., 2022). The important role of $O_3$ in the atmosphere has led
to many efforts focusing on $O_3$ observations that have improved our understanding of
atmospheric $O_3$ (Logan et al., 2012; Oetjen et al., 2016; Parrish et al., 2021). The limited spatial
coverage of $O_3$ observations promotes the efforts of spatial extensions of $O_3$ observations
(Chang et al., 2015; Peng et al., 2016). Recent advances in machine learning techniques further
provide a new method to extend $O_3$ observations by fusing satellite and surface observations
(Li et al., 2020; Liu et al., 2022; Wei et al., 2022).
Chemical transport models (CTMs), as powerful tools, have been widely used to simulate
and interpret observed $O_3$ variabilities (Parrington et al., 2012; Jiang et al., 2016; Li, Ke et al.,
2019). Despite the advances in CTMs, an accurate simulation of observed $O_3$ is still challenging
because of uncertainties in physical and chemical processes (Peng et al., 2021; Chen et al.,
2022), emission inventories (Elguindi et al., 2020; Jiang et al., 2022) and coarse model
resolutions (Schaap et al., 2015; Benavides et al., 2021). Furthermore, the high computational
cost is a bottleneck for rapid simulations, which poses a possible barrier to better understanding
tropospheric $O_3$. Alternatively, researchers may consider simulations of atmospheric $O_3$ with
the archived $O_3$ product and loss rates. For example, the tagged-$O_x$ mode of the GEOS-Chem
model has been used to analyze the sources and transport of tropospheric $O_3$ (Zhang et al., 2008;
Zhu et al., 2017; Han et al., 2018). However, it may not be an ideal choice to perform $O_3$
simulations based on the tagged-$O_x$ mode because $O_x$ is the combination of multiple species
($O_x=O_3+NO_2+2NO_3+3N_2O_5+HNO_3+HNO_4+PANs$) and thus cannot be accurately compared
with $O_3$ observations.

In this study, we developed the single $O_3$ tracer mode (tagged-$O_3$) of the GEOS-Chem

model, driven by archived $O_3$ product and loss rates provided by GEOS-Chem full chemistry
simulations, to build the capability of the GEOS-Chem model for rapid simulations of
tropospheric $O_3$ (rather than $O_x$). Data assimilations, by combining modeled and observed $O_3$
concentrations, can take advantage of both simulations and observations to produce more
accurate $O_3$ concentrations (Parrington et al., 2008; Ma et al., 2019; Huijnen et al., 2020). The
single $O_3$ tracer simulations were thus further combined with the Ozone Monitoring Instrument
(OMI) and China Ministry of Ecology and Environment (MEE) monitoring network $O_3$
observations (in this paper) and United States (US) Air Quality System (AQS) and European
AirBase network $O_3$ observations (in the companion paper, (Zhu et al., 2023)) via a sequential
Kalman Filter (KF) assimilation system (Tang et al., 2022; Han et al., 2022) to perform a
comparative analysis to investigate the changes in tropospheric $O_3$ in eastern (E.) China in
2015-2020 (in this paper) and the US and Europe in 2005-2020 (Zhu et al., 2023).

Satellite instruments provide globally covered $O_3$ observations that are sensitive to $O_3$

concentrations in the free troposphere. The OMI-based assimilations can thus reflect the
optimized adjustments in both global background and local $O_3$ concentrations. On the other
hand, surface observations are sensitive to local $O_3$ concentrations. Surface observation-based
assimilations can reflect the optimized adjustments in local contributions, and the information
of local contributions can be transported into the free troposphere via vertical convection in the
assimilation processes, which is different from the fusion of satellite and surface observations
(Li et al., 2020; Liu et al., 2022; Wei et al., 2022). Consequently, a comparative analysis by
assimilating satellite and surface $O_3$ observations is useful for better characterization of $O_3$
changes in the surface and free troposphere. Furthermore, the low computational costs of the
single O$_3$ tracer simulations allow us to design and perform different experiments much more
efficiently. Multiple simulation and assimilation experiments (see details in Table 1) were thus
conducted in this work to analyze the impacts of background O$_3$ (particularly, the interannual
and seasonal variabilities in the background O$_3$ as well as optimization in the background O$_3$)
and local O$_3$ formation on the changes in surface and free tropospheric O$_3$ over E. China.
This paper is organized as follows: in Section 2, we provide descriptions of the MEE and
OMI O$_3$ observations, the GEOS-Chem model and the single O$_3$ tracer simulation and
assimilation system used in this work. Tropospheric O$_3$ changes in E. China in 2015-2020 are
then demonstrated in Section 3 by assimilating MEE and OMI O$_3$ observations. As shown in
Fig. 1, five regions (i.e., North China Plain (#1), Yangtze River Delta (#2), Central China (#3),
Sichuan Basin (#4) and Southern China (#5)) are defined within the E. China domain. Regions
#1 and #2 are defined as highly polluted regions by excluding grids with low and medium
anthropogenic NO$_x$ emissions. Tropospheric O$_3$ changes over these regions are discussed to
investigate the possible regional discrepancies in surface and free tropospheric O$_3$ associated
with different local pollution levels. Our conclusions follow in Section 4.

**2. Data and Methods**
**2.1 Surface O$_3$ measurements**
We use MEE surface in situ O$_3$ concentration data (https://quotsoft.net/air/) for the period
2015-2020. These real-time monitoring stations report hourly concentrations of criteria
pollutants from 1691 sites in 2020. All stations (1441 urban sites and 250 urban background
sites) are assimilated in our analysis. Concentrations were reported by the MEE in µg m$^{-3}$ under
standard temperature (273 K) until 31 August 2018. This reference state was changed on 1
September 2018 to 298 K. We converted the O$_3$ concentrations to ppb and rescaled the post-
August 2018 concentrations to the standard temperature (273 K) to maintain consistency in the
trend analysis. It should be noted that the assimilation of $O_3$ observations from urban and urban
background sites may result in possible overestimation of surface $O_3$ concentrations over rural
areas.

**2.2 OMI PROFOZ product**

The OMI instrument was launched in July 2004 on the Aura spacecraft with a spatial
resolution of $13 \times 24$ km (nadir view). It provides globally covered measurements with
backscattered sunlight in the ultraviolet–visible range from 270 to 500 nm (UV1: 270–310 nm;
UV2: 310–365 nm; visible: 350–500 nm). In this study, we use the OMI $O_3$ profile retrieval
product (PROFOZ v0.9.3, level 2, Liu et al. (2010); Huang et al. (2017)) from the Smithsonian
Astrophysical Observatory (SAO). The retrieval uses the vector linearized discrete ordinate
radiative transfer model (VLIDORT) (Spurr, 2006) and Bayesian optimal estimation. Profiles
of partial $O_3$ columns (unit: DU) are retrieved in the spectral region of 270–330 nm with 24
vertical layers: approximately 2.5 km for each layer from the surface to approximately 60 km.
The following filters are applied in our analysis following Huang et al. (2017): 1) nearly clear-
sky scenes with effective cloud fraction $< 0.3$; 2) solar zenith angles (SZA) $< 75°$; and 3) fitting
root mean square (RMS, ratio of fitting residuals to assumed measurement error) $< 2.0$.
Starting in 2009, anomalies were found in the OMI data and diagnosed as attenuated
measured radiances in certain cross-track positions. This instrument degradation has been
referred to as the "row anomaly". To enhance the quality and stability of the data, only across-
track positions between 4-11 (within 30 positions in the UV1 channels) are assimilated in our
main assimilation experiment (Exp. #8). This treatment is similar to the production of row-
isolated data by using across-track positions between 3-18 (within 60 positions in the UV2
channels) in the OMI/MLS $O_3$ data (Ziemke et al., 2019; Wang, X. et al., 2022). The effects of
the usage of row-isolated data will be evaluated by comparing the main assimilation experiment
with the sensitivity assimilation experiment (Exp. #10) by assimilating OMI $O_3$ observations
at across-track positions 4-27.

The modeled tropospheric $O_3$ profiles in the assimilation processes and subsequent

analyses are convolved by using the OMI retrieval averaging kernels and a priori $O_3$ profile
based on the following equation (Liu et al., 2010; Huang et al., 2017):
$$\hat{x} = x_a + A(x - x_a) \qquad \text{(Eq. 1)}$$
where $\hat{x}$ is the modeled $O_3$ profile convolved by the retrieval averaging kernels, $x_a$ is the
OMI a priori $O_3$ profile, $x$ is the modeled $O_3$ profile, and $A$ is the OMI averaging kernel
matrix. Here $A(i, j) = \frac{\partial \hat{x}_j}{\partial x_i}$, representing the sensitivity of the retrieved partial $O_3$ column (DU)
at layer j to the change in $O_3$ (DU) at layer i. The unit for averaging kernels in this OMI product
is DU/DU because the conversion from DU to ppb varies with altitude.
**2.3 GEOS-Chem model configuration**

The GEOS-Chem chemical transport model (http://www.geos-chem.org, version 12-8-1)

is driven by assimilated meteorological data from MERRA-2. The GEOS-Chem full chemistry
simulation includes fully coupled $O_3$-$NO_x$-VOC-halogen-aerosol chemistry. Our analysis is
conducted at a horizontal resolution of nested 0.5°×0.625° over E. China with chemical
boundary conditions archived every 3 hours from global simulations with 4°×5° resolution.
Emissions are computed by the Harvard-NASA Emission Component (HEMCO). Global
default anthropogenic emissions are from the CEDS (Community Emissions Data System)
(Hoesly et al., 2018). Regional emissions are replaced by MEIC (Multiresolution Emission
Inventory for China) in China and MIX in other regions of Asia (Li et al., 2017). The reference
year for the CEDS inventory is 2010 with annual scaling factors in 2005-2014, and the
reference year for the MEIC/MIX inventory is 2010 with annual scaling factors in 2008-2010
in the GEOS-Chem model. Open fire emissions are from the Global Fire Emissions Database
(GFED4) (van der Werf et al., 2010).

Following Jiang et al. (2022), the total anthropogenic $NO_x$ and VOC emissions in the

GEOS-Chem model are scaled based on Zheng et al. (2018) and Li, M. et al. (2019) so that the

modeled surface nitrogen dioxide ($NO_2$) and $O_3$ concentrations in the a priori simulations are

identical to Jiang et al. (2022) in 2005-2018. The total anthropogenic $NO_x$ and VOC emissions

in 2019-2020 are further scaled based on linear projections. The total anthropogenic $NO_x$

emissions in the a priori simulations declined by 19% in China in 2015-2020. The total

anthropogenic VOC emissions in the a priori simulations increased by 1% in China in 2015-

2020. We refer the reader to Jiang et al. (2022) for the details of the model configuration and

performance, particularly the modeled trends of surface and tropospheric column $NO_2$ in 2005-

2018.

**2.4 Single $O_3$ tracer simulation**

A new chemical mechanism was developed in this work to allow the running of the single

$O_3$ tracer mode (tagged-$O_3$). As shown in Fig. S1 (see the SI), the package of the Kinetic

PreProcessor (KPP) module was modified to define the production ($PO_3$) and loss ($LO_3$) of $O_3$.

The GEOS-Chem full chemistry simulations with the updated KPP module were then

performed to produce $PO_3$ (unit kg $cm^{-3}$ $s^{-1}$) and relative $LO_3$ (i.e., $LO_3/[O_3]$ with unit $cm^{-3}$ $s^{-1}$)

every 20 minutes. Here the 20 minutes are selected to be the same as the chemical time step in

the GEOS-Chem full chemistry mode to ensure consistency between the single $O_3$ tracer and

full chemistry simulations. The single $O_3$ tracer simulation (tagged_o3_mod.F90) was then

performed by reading the archived $PO_3$ and relative $LO_3$. Because we are interested in

tropospheric chemistry, we archived $O_3$ concentrations instead of $O_3$ production and loss rates

in the stratosphere in the full chemistry simulations. The archived stratospheric $O_3$

concentrations were read in the single $O_3$ tracer simulation process as boundary conditions to

ensure a reasonable stratospheric-tropospheric $O_3$ exchange.

The major advantage of the single $O_3$ tracer simulation is dramatic reductions in

computational costs by approximately 91%-94%; for example, the computational costs (hours
of wall time for one year simulation) are 57.5 and 5.2 hours at the global scale (4°×5°) and 80.2
and 4.5 hours within the nested China domain (0.5°×0.625°) by full chemistry and single $O_3$
tracer simulations, respectively. Consequently, once $PO_3$ and $LO_3$ are produced, the
computational costs of performing additional single $O_3$ tracer simulations are almost negligible.
The low computational costs of the single $O_3$ tracer simulation allow us to design and perform
different simulation and assimilation experiments much more efficiently. As shown in Table
1, there are 10 different simulation and assimilation experiments performed in this work, which
requires 4812 hours (wall time) with the full chemistry simulation but only 270 hours (wall
time) with the single $O_3$ tracer simulation.
Here we evaluate the consistency in modeled $O_3$ concentrations between single $O_3$ tracer
and full chemistry simulations. Fig. 2A1-A5 show the annual and seasonal averages of the
surface maximum daily 8-hour average (MDA8) $O_3$ over E. China in 2015-2020 from the full
chemistry simulation. The modeled surface MDA8 $O_3$ concentrations are as high as 60-70 ppb
in the summer and as low as 10-20 ppb in the winter over northern China. The simulation with
the single $O_3$ tracer mode (Fig. 2B1-B5) demonstrates spatial consistency with the full
chemistry simulation (Fig. 2A1-A5) and temporal consistency at both the daily (Fig. 3A) and
monthly (Fig. 3B) scales in 2015-2020. In contrast, the tagged-$O_x$ mode of the GEOS-Chem
model is driven by the archived production and loss of $O_x$, which is the combination of multiple
species including $O_3$. There are large discrepancies between full chemistry (Fig. 2A1-A5) and
tagged-$O_x$ (Fig. 2C1-C5) simulations. As shown in Fig. 3, the $O_x$ concentrations are higher than
the $O_3$ concentrations by approximately 6 ppb, and the relative difference can reach 40% in the
winter. Our analysis thus indicates the reliability of the single $O_3$ tracer simulations developed
in this work.
**2.5 Data assimilation method**

We employ the sequential KF to assimilate $O_3$ observations, which has been used in

recent studies to optimize tropospheric CO concentrations (Tang et al., 2022; Han et al., 2022).
As a brief description of the assimilation algorithm, the forward model (**M**) predicts the $O_3$
concentration ($\boldsymbol{x}_{at}$) at time t:
$$\boldsymbol{x}_{at} = \mathbf{M}_t \boldsymbol{x}_{t-1} \quad (\text{Eq. 2})$$
The optimized $O_3$ concentrations can be expressed as:
$$\boldsymbol{x}_t = \boldsymbol{x}_{at} + \mathbf{G}_t(\boldsymbol{y}_t - \mathbf{K}_t \boldsymbol{x}_{at}) \quad (\text{Eq. 3})$$
where $\boldsymbol{y}_t$ is the observation and $\mathbf{K}_t$ represents the operation operator that projects $O_3$
concentrations from the model space to the observation space. $\mathbf{G}_t$ is the KF gain matrix, which
can be described as:
$$\mathbf{G}_t = \mathbf{S}_{at}\mathbf{K}_t^T(\mathbf{K}_t\mathbf{S}_{at}\mathbf{K}_t^T + \mathbf{S}_\epsilon)^{-1} \quad (\text{Eq. 4})$$
where $\mathbf{S}_{at}$ and $\mathbf{S}_\epsilon$ are the model and observation covariances, respectively. The optimized $O_3$
concentrations provided by Eq. 3 are then forwarded (hourly) to Eq. 2. The model errors are
assumed to be 50% because the objective of our assimilations is to provide dynamic extensions
of atmospheric $O_3$ observations. The a posteriori $O_3$ concentrations with the assumption of 50%
model errors are expected to match better with atmospheric $O_3$ observations. The measurement
errors are calculated as $\varepsilon_0 = ermax + 0.0075 * \Pi_0$, where *ermax* is the base error (1.5 µg m⁻
³) and $\Pi_0$ represents the observed $O_3$ concentrations (unit: µg m⁻³). The representation errors
are calculated as $\varepsilon_r = \gamma\varepsilon_0\sqrt{\Delta l/L}$, where $\gamma$ is a scaling factor (0.5), $\Delta l$ is the model resolution
(~56 km in this study), and L represents the range that the observation can reflect, which
depends on the station type (2 km for urban, 4 km for suburban). The total observation error is
then defined as $\varepsilon_t = \sqrt{\varepsilon_0^2 + \varepsilon_r^2}$. Furthermore, the "superobservation" method was applied in
this work to further reduce the influence of representative error (Miyazaki et al., 2017; Tang et
al., 2022):
$$\omega_j = 1/\varepsilon_j^2 \quad (\text{Eq. 5})$$
$$y_s = \sum_{j=1}^{k} \omega_j y_j / \sum_{j=1}^{k} \omega_j \qquad \text{(Eq. 6)}$$
$$1/\varepsilon_s^2 = \sum_{j=1}^{k} 1/\varepsilon_j^2 \qquad \text{(Eq. 7)}$$
where $y_j$ is O₃ observation of the $j$th station, $\omega_j$ represents the weighting factor of the $j$th
station, $y_s$ and $\varepsilon_s$ are the grid-based O₃ observations and errors (superobservation),
respectively.

## 3. Results and Discussion

### 3.1 Surface O₃ by assimilating MEE O₃ observations

We first investigate the effects of surface O₃ observations on single O₃ tracer
assimilations. O₃ at the surface level is formed by precursors mixed in the planetary boundary
layer (PBL). Thus, it may not be accurate to assume that the differences between simulated and
observed surface O₃ concentrations are completely caused by biased O₃ production and loss at
the surface level. Here we adjust O₃ concentrations above the surface level within the PBL
when assimilating surface O₃ observations:
$$\Delta O_3^n = \Delta O_3^1 \times \gamma^{n-1} \qquad \text{(Eq. 8)}$$
where $\Delta O_3^1$ is the adjustment at the surface level calculated with Eq. 3; $\Delta O_3^n$ is the adjustment
at model level $n$, which is based on $\Delta O_3^1$ but decays exponentially with the increase in model
level, and the decay speed is adjusted by the $\gamma$ parameter. As shown in Table 1, three
assimilation experiments (Exp. #5-#7) were conducted to evaluate the effects of the decay
speed: 1) $\gamma = 0$ by assuming that the biased surface O₃ concentrations are completely caused
by biased O₃ production and loss at the surface level; 2) $\gamma = 1$ by assuming full mixing of O₃
biases within the PBL; and 3) $\gamma = 0.8$ by assuming partial mixing of O₃ biases within the
PBL, i.e., the adjustment at the 4th model level is approximately 50% of $\Delta O_3^1$, and the
adjustment at the 10th model level (close to the top of PBL) is approximately 10% of $\Delta O_3^1$.
As shown in Fig. S2A (see the SI), the assimilated surface MDA8 $O_3$ concentrations
show good agreement by using different $\gamma$ parameters: 42.3, 41.8 and 42.0 ppb ($\gamma = 0$, 0.8 and
1.0) in 2015-2020; there are noticeable discrepancies in the trends of assimilated surface $O_3$
concentrations: 0.80, 1.24 and 1.50 ppb $yr^{-1}$ ($\gamma = 0$, 0.8 and 1.0) in 2015-2020 (Fig. S2B), and
the trends obtained by considering the mixing of $O_3$ biases ($\gamma = 0.8$ and 1.0) match better with
MEE $O_3$ observations (1.77 ppb $yr^{-1}$). Fig. S3 (see the SI) further demonstrates tropospheric $O_3$
columns by assimilating MEE $O_3$ observations in 2015-2020. We find good agreement in the
assimilated tropospheric $O_3$ columns by using different $\gamma$ parameters, i.e., the mean
tropospheric $O_3$ columns are 38.1, 37.9 and 37.9 DU, and the trends of tropospheric $O_3$ columns
are 0.11, 0.17 and 0.21 ppb $yr^{-1}$ ($\gamma = 0$, 0.8 and 1.0). Considering the better agreement in the
trends of assimilated surface $O_3$ concentrations ($\gamma = 0.8$ and 1.0) with observations, we finally
decide to set $\gamma = 0.8$ as our main assimilation setting by assuming partial mixing of $O_3$ biases
within the PBL.
Fig. 4A1-A5 show the annual and seasonal averages of surface MDA8 $O_3$ observations
from MEE stations in 2015-2020. Fig. 4C1-C5 show the annual and seasonal averages of the a
posteriori $O_3$ concentrations by assimilating the MEE $O_3$ observations. As shown in Fig. 5, the
assimilated $O_3$ concentrations (blue lines) show good agreement with MEE $O_3$ observations
(red lines): the mean surface MDA8 $O_3$ in 2015-2020 are 43.2, 41.8 and 42.1 ppb (E. China),
42.4, 45.6 and 47.6 ppb (North China Plain), 44.6, 45.0 and 44.9 ppb (Yangtze River Delta),
45.1, 43.1 and 43.5 ppb (Central China), 45.7, 37.5 and 36.9 ppb (Sichuan Basin), and 43.2,
39.2 and 38.3 ppb (Southern China) in the a priori simulations, a posteriori simulations and
MEE observations, respectively. It should be noted that Fig. 5A exhibits broadly good
agreement between the a priori and a posteriori $O_3$ concentrations over E. China except for a
larger difference in the summer. However, as shown in Fig. 4D1-D5, the good agreements
between the a priori and a posteriori $O_3$ concentrations are caused by the counterbalance of
positive biases (i.e., overestimated surface $O_3$ in the a priori simulations over southern China)
and negative biases (i.e., underestimated surface $O_3$ in the a priori simulations over northern
China). The good agreements in Fig. 5A thus cannot represent good performance in the
simulations of surface $O_3$ concentrations.
The assimilations exhibit noticeable declines in surface $O_3$ concentrations over regions
#2-5 in June-July, and the declines are underestimated by the a priori simulations (Fig. 5C-F).
The inaccurate simulation in June-July thus results in overestimated surface $O_3$ concentrations
in the summer. There is dramatic seasonality in surface $O_3$ concentrations (Fig. 5): maximum
in June in the North China Plain, May and August in the Yangtze River Delta, Central China
and Sichuan Basin, and September-October in Southern China. Fig. 4E1-E5 exhibits the effects
of seasonal variabilities in background $O_3$ (Exp. #3) by fixing background $O_3$ in the spring in
the simulations. The fixed background $O_3$ has limited influences on surface $O_3$ concentrations,
and consequently, the seasonality in surface $O_3$ concentrations is dominated by local
contributions. As we expected, MDA8 $O_3$ concentrations are higher over areas with higher
anthropogenic $NO_x$ emissions, for example, 45.6 and 45.0 ppb in the North China Plain and
Yangtze River Delta, respectively, in contrast to 43.1, 37.5 and 39.2 ppb in Central China,
Sichuan Basin and Southern China. The influences of regional transport on surface $O_3$
concentrations are limited; for example, $O_3$ generated within the North China Plain PBL by
setting $O_3$ formation rates within the North China Plain PBL to zero (Exp. #4) are mainly
contained within the North China Plain (Fig. 4F1-F5).
**3.2 Rapid increasing trends in surface $O_3$ concentrations**
Here we investigate the changes in surface $O_3$ concentrations from observations and
assimilations. As shown in Fig. 6B1-B5, the a priori simulation suggests slightly increasing
trends of MDA8 $O_3$ in 2015-2020: 0.31 (spring), -0.12 (summer), 0.45 (autumn) and 0.40
(winter) ppb $yr^{-1}$, and the relative increasing trends are 0.7 (spring), -0.2 (summer), 1.1
(autumn) and 1.4 (winter) % $yr^{-1}$. The a priori simulation suggests increasing trends of surface
$O_3$ concentrations in the summer over areas with higher local pollution levels, for example,
0.68 and 0.63 ppb $yr^{-1}$ over the North China Plain and Yangtze River Delta, respectively, and
decreasing trends of surface $O_3$ concentrations in the summer over areas with lower local
pollution levels, for example, -0.83 and -1.01 ppb $yr^{-1}$ over the Sichuan Basin and Southern
China, respectively. The decreasing trends over areas with lower local pollution levels in the
simulations are not surprising, given the decreases in anthropogenic $NO_x$ emissions (Zheng et
al., 2018; Jiang et al., 2022) and the reported $NO_x$-limited $O_3$ nonlinear chemical regimes in
model simulations (Chen et al., 2021; Liu et al., 2021). Furthermore, as shown in Fig. 6D1-D5,
the interannual variabilities in background $O_3$ (Exp. #2) are suggested to result in increases in
surface $O_3$ concentrations in the a priori simulations in 2015-2020 by 0.02 (spring), 0.05
(summer), 0.02 (autumn) and 0.00 (winter) ppb $yr^{-1}$, and the relative contribution is particularly
pronounced in the summer.

In contrast, the increasing trends in surface $O_3$ are much stronger in the assimilations. As

shown in Table 2.1, our assimilation suggests 1.60 (spring), 1.16 (summer), 1.47 (autumn) and
0.80 (winter) ppb $yr^{-1}$ increases in surface $O_3$ over E. China in 2015-2020, and the relative
increasing trends are 3.4 (spring), 2.2 (summer), 3.7 (autumn) and 2.7 (winter) % $yr^{-1}$. The
annual increasing trend (1.24 ppb $yr^{-1}$) in the assimilated surface $O_3$ concentrations is more
consistent with the MEE $O_3$ observations (1.77 ppb $yr^{-1}$) which are comparable with the
reported recent trends in surface $O_3$ concentrations in China of 1.25-2.0 ppb $yr^{-1}$
(Mousavinezhad et al., 2021; Wei et al., 2022; Wang, W. et al., 2022). The increasing trends
are weaker when the modeled surface $O_3$ concentrations are averaged over E. China (Table
2.2) instead of sampling at the locations and times of MEE observations: 0.71 (spring), 0.36
(summer), 0.69 (autumn) and 0.54 (winter) ppb $yr^{-1}$ because most MEE stations are urban sites.
Our analysis thus indicates a noticeable underestimation in the increasing trends of surface $O_3$
concentrations in China in the a priori simulations, particularly in the summer, despite the
anthropogenic $NO_x$ and VOC emissions having been scaled in the simulations following Jiang
et al. (2022).

The changes in surface $O_3$ concentrations have significant regional and seasonal

discrepancies. As shown in Tables S1-S5 (see the SI), our assimilations demonstrate strong
increasing trends in surface $O_3$ concentrations in 2015-2020 in spring (1.94 ppb $yr^{-1}$ or 3.8%
$yr^{-1}$) and summer (2.52 ppb $yr^{-1}$ or 4.0% $yr^{-1}$) over the North China Plain; in spring (2.21 ppb
$yr^{-1}$ or 4.4% $yr^{-1}$) and autumn (1.84 ppb $yr^{-1}$ or 4.1% $yr^{-1}$) over the Yangtze River Delta; in
spring (2.07 ppb $yr^{-1}$ or 4.3% $yr^{-1}$) and autumn (2.09 ppb $yr^{-1}$ or 4.7% $yr^{-1}$) over Central China;
in spring (1.69 ppb $yr^{-1}$ or 3.8% $yr^{-1}$) over the Sichuan Basin; and in autumn (2.21 ppb $yr^{-1}$ or
4.9% $yr^{-1}$) over Southern China. While surface $O_3$ concentrations are higher over areas with
higher anthropogenic $NO_x$ emissions, the increasing trends in surface $O_3$ concentrations over
Central China and Southern China are comparable with those in the North China Plain and
Yangtze River Delta. Our analysis advises more attention to $O_3$ pollution in spring and autumn
over areas with lower anthropogenic $NO_x$ emissions because of the rapid increases in surface
$O_3$ concentrations.
**3.3 Tropospheric $O_3$ columns by assimilating OMI $O_3$ observations**

Fig. 7A1-A5 show the annual and seasonal averages of tropospheric OMI $O_3$ columns in

2015-2020. OMI is sensitive to $O_3$ at different vertical levels (Huang et al., 2017; Fu et al.,
2018), and thus, the standard KF algorithm (Eq. 3) was employed to adjust tropospheric $O_3$
vertical profiles with the application of OMI $O_3$ averaging kernels. Fig. 7C1-C5 show the
annual and seasonal averages of the a posteriori tropospheric $O_3$ columns by assimilating OMI
$O_3$ observations. The assimilated tropospheric $O_3$ columns show good agreement with OMI $O_3$
observations: the mean tropospheric $O_3$ columns in 2015-2020 (Table 2.3) are 37.1 DU in the
a priori simulations and 37.9 and 38.0 DU in the a posteriori simulation and OMI observations,

respectively. The discrepancies between the a priori and a posteriori simulations in tropospheric $O_3$ columns (Fig. 7) are smaller than those in surface $O_3$ concentrations (Fig. 4). A better simulation capability in tropospheric column $O_3$ is expected because model simulation with $0.5°\times0.625°$ horizontal resolution may not be enough to accurately resolve $O_3$ nonlinear chemical regimes over urban surface stations.

The above assimilated tropospheric $O_3$ columns (Exp. #8) are driven by optimized $O_3$ background conditions provided by global assimilations of OMI $O_3$ as well as row-isolated OMI data by using across-track positions between 4-11. Fig. 7E1-E5 exhibits the effects of optimization on regional $O_3$ background conditions. The mean assimilated tropospheric $O_3$ column driven by the original $O_3$ background conditions is 37.6 DU (Exp. #9), which is slightly lower than the 37.9 DU in the main assimilation (Exp. #8). The usage of original $O_3$ background conditions can result in overestimations over southern China in the spring and winter, and underestimations over northern China in the spring and summer (Fig. 7E1-E5). Fig. 7F1-F5 further exhibits the effects of the usage of row-isolated data. The mean assimilated tropospheric $O_3$ column by assimilating OMI $O_3$ observations at across-track positions 4-27 is 37.7 DU (Exp. #10), which is slightly lower than the 37.9 DU in the main assimilation (Exp. #8). The underestimations in the assimilated tropospheric $O_3$ columns are particularly significant in the spring and summer (Fig. 7F2-F3).

As shown in Fig. 8, the trends of tropospheric $O_3$ columns in 2015-2020 (Table 2.3) are 0.02 DU $yr^{-1}$ in the a priori simulations and -0.17 and -0.30 DU $yr^{-1}$ in the a posteriori simulation and OMI observations, respectively. In contrast to the wide distributions of increasing trends of $O_3$ at the surface level (Fig. 6), both OMI $O_3$ observations (-0.30 DU $yr^{-1}$) and the OMI-based assimilations (-0.17 DU $yr^{-1}$) suggest decreasing trends in tropospheric $O_3$ columns over E. Asia in 2015-2020 (Fig. 8). The decreasing trends are stronger in the summer and weaker in the spring. Furthermore, the usage of original $O_3$ background conditions can

result in overestimated trend by approximately 0.08 DU yr$^{-1}$ (Fig. 8D1); and the assimilation
of OMI $O_3$ observations at across-track positions 4-27 can result in a similar overestimated
trend, by approximately 0.08 DU yr$^{-1}$ (Fig. 8E1). These discrepancies demonstrate the
importance of optimized usages of regional $O_3$ background conditions and OMI $O_3$
observations in the assimilations.
**3.4 Changes in tropospheric $O_3$ columns**

The trends shown in Fig. 8 may not represent the actual tropospheric $O_3$ changes well

because the convolution of OMI $O_3$ averaging kernels on the output $O_3$ profiles can affect the
weights of the derived tropospheric columns to $O_3$ at different vertical levels. Consequently,
Fig. 9 shows the annual and seasonal averages of tropospheric $O_3$ columns in which the output
$O_3$ profiles are not convolved with OMI retrieval averaging kernels so that they can better
represent the actual atmospheric $O_3$ state. The assimilated tropospheric $O_3$ columns are 37.9
and 38.8 DU (E. China), 42.9 and 43.7 DU (North China Plain), 47.5 and 48.1 DU (Yangtze
River Delta), 47.4 and 48.1 DU (Central China), 43.8 and 44.6 DU (Sichuan Basin), and 39.6
and 40.6 DU (Southern China) in 2015-2020 by assimilating MEE and OMI $O_3$ observations,
respectively.

In contrast to the higher surface MDA8 $O_3$ concentrations over areas with higher

anthropogenic $NO_x$ emissions, tropospheric $O_3$ columns over Central China and the Sichuan
Basin are even higher than those over the highly polluted North China Plain. In addition,
tropospheric $O_3$ columns obtained by assimilating MEE surface $O_3$ observations are lower than
those obtained by assimilating OMI $O_3$ observations, and their difference is larger in the
summer and smaller in the winter. As shown in Fig. S4 (see the SI), the impacts of different
surface and satellite $O_3$ observations on the assimilated $O_3$ vertical profiles are limited. The
assimilation of MEE surface $O_3$ observations leads to decreases in $O_3$ concentrations in the
lower troposphere from the surface to 600 hPa levels over the Sichuan Basin and Southern
China; the assimilation of OMI $O_3$ observations leads to enhancement in $O_3$ concentrations in
the middle and upper troposphere over the highly polluted North China Plain.

The assimilated tropospheric $O_3$ columns are maximum in June-July over the highly

polluted North China Plain and March-May over other lower polluted regions (Fig. S5, see the
SI). Fig. 9E1-E5 exhibit the effects of seasonal variabilities in background $O_3$ (Exp. #3). The
fixed background $O_3$ in the spring can result in dramatic increases in tropospheric $O_3$ columns
by 14.3 (summer), 15.1 (autumn) and 4.8 (winter) DU over E. China. Fig. 9F1-F5 further
exhibit the effects of $O_3$ formation within the North China Plain PBL (Exp. #4) on tropospheric
$O_3$ columns, which are 5.4 (spring), 8.1 (summer), 3.6 (autumn) and 1.3 (winter) DU over the
North China Plain. In addition, as shown in Fig. S6 (see the SI), there is a larger enhancement
in $O_3$ production rates in the free troposphere (600-300 hPa) over the North China Plain in the
summer than in other lower polluted regions. Consequently, the spring maximum in
tropospheric $O_3$ columns over lower polluted regions is caused by the enhanced background $O_3$
(Fig. 9E1-E5), and the summer maximum in tropospheric $O_3$ columns over the highly polluted
North China Plain is caused by the local contributions from enhanced $O_3$ formation within the
North China Plain PBL (Fig. 9F1-F5) and free troposphere (Fig. S6).

As shown in Fig. 10A1-A5, the trends of tropospheric $O_3$ columns in the a priori

simulations in 2015-2020 are -0.02 (spring), 0.02 (summer), 0.29 (autumn) and 0.09 (winter)
DU $yr^{-1}$ over E. China. The interannual variability in background $O_3$ (Fig. 10D1-D5, Exp. #2)
is suggested to have important contributions to the trends of tropospheric $O_3$ columns by 0.09
(spring), -0.11 (summer), -0.10 (autumn) and -0.08 (winter) DU $yr^{-1}$. The trends of assimilated
tropospheric $O_3$ columns are 0.17 and -0.10 DU $yr^{-1}$ (E. China), which are comparable with the
reported recent trend in free tropospheric $O_3$ concentrations over China by -0.14 DU $yr^{-1}$
(Dufour et al., 2021), and are 0.47 and 0.12 DU $yr^{-1}$ (North China Plain), 0.45 and 0.13 DU $yr^{-}$
$^1$ (Yangtze River Delta), 0.32 and -0.06 DU $yr^{-1}$ (Central China), 0.03 and -0.29 DU $yr^{-1}$
(Sichuan Basin), and 0.06 and -0.25 DU yr$^{-1}$ (Southern China) by assimilating MEE and OMI
O$_3$ observations, respectively.

The stronger increasing trends in tropospheric O$_3$ columns over the highly polluted North

China Plain (Fig. 10A1) are suggested to be caused by larger local contributions because of
relatively uniform influences from interannual variability in background O$_3$ (Fig. 10D1).
Higher positive trends by assimilating MEE observations are expected, given the increasing
trends in surface O$_3$ concentrations (1.77 ppb yr$^{-1}$) and decreasing trends in OMI O$_3$
concentrations (-0.30 DU yr$^{-1}$) over E. China. Furthermore, it should be noted that while the
Yangtze River Delta is defined as a highly polluted region in our analysis, its area is much
smaller than that of the North China Plain (Fig. 1); thus, the impact of local contributions on
tropospheric O$_3$ columns over the Yangtze River Delta is not as strong as that over the North
China Plain.
**4. Conclusion**

A single O$_3$ tracer (tagged-O$_3$) mode was developed in this work to build the capability

of the GEOS-Chem model for rapid simulations of tropospheric O$_3$. The single O$_3$ tracer
simulation demonstrates consistency with the GEOS-Chem full chemistry simulation. In
contrast, the O$_x$ concentrations provided by the tagged-O$_x$ mode are higher than the O$_3$
concentrations by approximately 6 ppb, and the relative difference can reach 40% in the winter,
which is thus not suitable for direct comparison with observed O$_3$. The computational costs of
the single O$_3$ tracer mode are reduced by approximately 91-94% with respect to the full
chemistry mode. For example, the computational costs (hours of wall time per simulation year)
are 57.5 and 5.2 hours at the global scale (4°×5°) and 80.2 and 4.5 hours within the nested
China domain (0.5°×0.625°) by full chemistry and single O$_3$ tracer simulations, respectively.
The low computational costs allow us to design and perform different experiments much more
efficiently. As shown in Table 1, 10 different simulation and assimilation experiments are
performed in this work to analyze the impacts of background and local contributions to surface
and free tropospheric $O_3$ changes over E. China in 2015-2020, which requires 4812 hours (wall
time) with the full chemistry simulation but only 270 hours (wall time) with the single $O_3$ tracer
simulation.

As an application of the single $O_3$ tracer mode, the assimilated surface $O_3$ concentrations

demonstrate good agreement with surface $O_3$ observations: 43.2, 41.8 and 42.1 ppb over E.
China in a priori and a posteriori simulations and observations, respectively. We find noticeable
biases in modeled surface $O_3$ concentrations, for example, overestimated surface $O_3$ over
southern China and underestimated surface $O_3$ over northern China. The assimilations indicate
rapidly increasing trends in surface $O_3$ concentrations by 1.60 (spring), 1.16 (summer), 1.47
(autumn) and 0.80 (winter) ppb $yr^{-1}$ over E. China in 2015-2020, and the increasing trends are
underestimated by the a priori simulations. While surface $O_3$ concentrations are higher over
areas with higher anthropogenic $NO_x$ emissions, the increasing trends in surface $O_3$
concentrations over Central China and Southern China are comparable with those in the North
China Plain and Yangtze River Delta. Our analysis thus advises more attention to $O_3$ pollution
in spring and autumn over areas with lower anthropogenic $NO_x$ emissions in China because of
the rapid increases in surface $O_3$ concentrations.

Similarly, the assimilated tropospheric $O_3$ columns demonstrate good agreement with

OMI observations: 37.1, 37.9 and 38.0 DU over E. China in a priori and a posteriori simulations
(convolved with OMI retrieval averaging kernels) and OMI observations, respectively. The
trends of assimilated tropospheric $O_3$ columns in 2015-2020 over E. China are 0.09 and -0.17
(spring), 0.17 and -0.22 (summer), 0.38 and 0.04 (autumn), and 0.12 and -0.02 (winter) by
assimilating MEE and OMI $O_3$ observations, respectively. We find stronger increasing trends
in tropospheric $O_3$ columns over highly polluted areas due to the larger local contributions, for
example, 0.47 and 0.12 DU $yr^{-1}$ (North China Plain) in contrast to 0.03 and -0.29 DU $yr^{-1}$
(Sichuan Basin) and 0.06 and -0.25 DU $yr^{-1}$ (Southern China) by assimilating MEE and OMI
$O_3$ observations, respectively. The large discrepancy by assimilating surface and satellite
observations indicates the possible uncertainties in the derived free tropospheric $O_3$ changes.
The usage of optimized $O_3$ background conditions and row-isolated OMI data is important to
produce more reliable results, for example, the usage of original $O_3$ background conditions can
result in an overestimated trend by approximately 0.08 DU $yr^{-1}$ in 2015-2020.

Our analysis demonstrates noticeable contributions of the interannual variability in

background $O_3$ to the trends in tropospheric $O_3$ over E. China. The seasonality in surface $O_3$
concentrations is dominated by local contributions; however, the interannual variabilities in
background $O_3$ have noticeable contributions to the increasing trends in surface $O_3$ particularly
in the summer in the a priori simulations. Moreover, the spring maximum in tropospheric $O_3$
columns over lower polluted regions is caused by the enhanced background $O_3$, and the summer
maximum in tropospheric $O_3$ columns over the highly polluted North China Plain is caused by
enhanced local $O_3$ formation. The interannual variabilities in background $O_3$ have important
contributions to the trends in tropospheric $O_3$ columns; for example, the trends of tropospheric
$O_3$ columns in 2015-2020 are -0.02 (spring), 0.02 (summer), 0.29 (autumn) and 0.09 (winter)
DU $yr^{-1}$ over E. China, and the contributions from interannual variability in background $O_3$ are
0.09 (spring), -0.11 (summer), -0.10 (autumn) and -0.08 (winter) DU $yr^{-1}$ in the a priori
simulations. Our analysis thus suggests more attention to the impact of background $O_3$ to
tropospheric O3 changes in China, particularly in the free troposphere.

The capability of rapid $O_3$ simulation developed in this work is a useful tool for

interpreting atmospheric $O_3$ observations. Assimilations of surface and satellite observations,
as shown in this work, can provide useful information to better describe the changes in surface
and free tropospheric $O_3$. Despite these advantages, it should be noted that the linear chemistry
assumption by reading the archived $PO_3$ and $LO_3$ implies single $O_3$ tracer mode is good for
representing near-current $O_3$ chemical conditions, particularly, for scientific issues associated
with the sources and transport of tropospheric $O_3$ as well as assimilations in this work and the
companion paper (Zhu et al., 2023). More cautious applications are suggested under
substantially different $O_3$ chemical conditions as the linear chemistry assumption could not be
satisfied.

**Code and data availability:** The MEE $O_3$ data can be downloaded from
https://quotsoft.net/air/. The AQS and AirBase surface $O_3$ data can be downloaded from
https://www.eea.europa.eu/data-and-maps/data/aqereporting-8 and
https://aqs.epa.gov/aqsweb/airdata/download_files.html#Row. The OMI PROFOZ product
can be acquired at
https://avdc.gsfc.nasa.gov/pub/data/satellite/Aura/OMI/V03/L2/OMPROFOZ/. The GEOS-
Chem model (version 12.8.1) can be downloaded from http://wiki.seas.harvard.edu/geos-
chem/index.php/GEOS-Chem_12#12.8.1. The KPP module for tagged-$O_3$ simulations can be
downloaded from https://doi.org/10.5281/zenodo.7545944.

**Author Contributions**: Z.J. designed the research. R.Z. developed the model code and
performed the research. Z.J. and R.Z. wrote the manuscript. X.L. provided instruction for the
usage of OMI data. All authors contributed to discussions and editing the manuscript.

**Competing interests**: The contact author has declared that neither they nor their co-authors
have any competing interests.

**Acknowledgments:** We thank the China Ministry of Ecology and Environment (MEE), the
United States Environmental Protection Agency and the European Environmental Agency for
providing the surface $O_3$ measurements. The numerical calculations in this paper have been
done on the supercomputing system in the Supercomputing Center of University of Science
and Technology of China. This work was supported by the Hundred Talents Program of
Chinese Academy of Science and National Natural Science Foundation of China (42277082,

41721002).

## Table and Figures


**Table 1.** Single $O_3$ tracer simulation and assimilation experiments (Exp.) conducted in this
work. Exp. #1: the main a priori simulation; Exp. #2: $O_3$ boundary conditions and stratospheric
$O_3$ concentrations are fixed in 2015; Exp. #3: $O_3$ boundary conditions and stratospheric $O_3$
concentrations are fixed in the spring; Exp. #4: $O_3$ formation rates within the North China Plain
PBL are set to zero; Exp. #5: the main assimilation by assimilating MEE surface $O_3$
observations with $\gamma = 0.8$; Exp. #6: only surface $O_3$ concentrations are adjusted ($\gamma = 0$); Exp.
#7: full mixing of $O_3$ biases within the PBL ($\gamma = 1.0$); Exp. #8: the main assimilation by
assimilating OMI $O_3$ observations; Exp. #9: $O_3$ boundary conditions are not optimized; Exp.
#10: assimilating OMI $O_3$ observations at across-track positions 4-27.

**Table 2.** Averages (with units ppb or DU) and trends (with units ppb $yr^{-1}$ or DU $yr^{-1}$) of surface
and tropospheric column $O_3$ concentrations in 2015-2020 over E. China from observations
(MEE and OMI) and a priori (Exp. #1) and a posteriori (KF) simulations (Exp. #5 and #8). The
domain definition of E. China is shown by Fig. 1A. T2.1): the modeled surface $O_3$ is sampled
at the locations and times of MEE surface $O_3$ observations; T2.2): the modeled surface $O_3$ is
averaged over E. China (land only); T2.3): the output $O_3$ profiles from the a priori and a
posteriori simulations are convolved with OMI $O_3$ averaging kernels; T2.4): the output $O_3$
profiles are NOT convolved with OMI $O_3$ averaging kernels. The uncertainties in the averages
are calculated using the bootstrapping method. The trends and uncertainties in the trends are
calculated using the linear fitting of averages by using the least squares method (see details in
the SI).

**Fig. 1.** (A) Anthropogenic $NO_x$ emissions over E. China in 2015; (B) Region definitions for
the North China Plain (#1), Yangtze River Delta (#2), Central China (#3), Sichuan Basin (#4)
and Southern China (#5). The different colors (red, gray and green) represent grids with high
(highest 15%), medium (15-50%) and low (lowest 50%) anthropogenic $NO_x$ emissions.
Regions #1 and #2 are defined as highly polluted (HP) regions by excluding grids with low and
medium anthropogenic $NO_x$ emissions.

**Fig. 2.** Surface MDA8 $O_3$ in 2015-2020 (annual and seasonal averages) simulated by GEOS-
Chem model with (A1-A5) full chemistry mode; (B1-B5) single $O_3$ tracer (tagged-$O_3$) mode;
and (C1-C5) tagged-$O_x$ mode. The 8-hour range of surface $O_x$ is selected according to the time
range of MDA8 $O_3$.

**Fig. 3.** (A) Daily averages of surface MDA8 $O_3$ over E. China in 2015-2020 from GEOS-Chem
full chemistry (black), single $O_3$ tracer (tagged-$O_3$) (blue) and tagged-$O_x$ (red) simulations; (B)
Monthly averages of MDA8 $O_3$. The dashed lines in panel B are annual averages.

**Fig. 4.** Surface MDA8 $O_3$ in 2015-2020 (annual and seasonal averages) from (A1-A5) MEE
stations; (B1-B5) GEOS-Chem a priori simulation (Exp. #1); (C1-C5) GEOS-Chem a
posteriori simulation by assimilating MEE $O_3$ observations (Exp. #5); (D1-D5) Bias in the a
priori simulations (Exp. #1 minus #5). (E1-E5) Effects of seasonal variabilities in background
$O_3$ (Exp. #3 minus #1); (F1-F5) Effects of $O_3$ formation within the North China Plain PBL
(Exp. #1 minus #4).

**Fig. 5.** (A-F) Daily averages of surface MDA8 $O_3$ in 2015-2020 from MEE stations (red) and
GEOS-Chem a priori (black, Exp. #1) and a posteriori (blue, Exp. #5) simulations by
assimilating MEE $O_3$ observations. (G-L) Monthly averages of MDA8 $O_3$. The dashed lines in
panels G-L are annual averages. The domain definition of E. China is shown by Fig. 1A.

**Fig. 6.** Trends of surface MDA8 $O_3$ in 2015-2020 (annual and seasonal averages) from (A1-
A5) MEE stations; (B1-B5) GEOS-Chem a priori simulation (Exp. #1); (C1-C5) GEOS-Chem
a posteriori simulation by assimilating MEE $O_3$ observations (Exp. #5). (D1-D5) Effects of
interannual variabilities in background $O_3$ (Exp. #1 minus #2).

**Fig. 7.** Tropospheric $O_3$ columns in 2015-2020 (annual and seasonal averages) from (A1-A5)
OMI observations; (B1-B5) GEOS-Chem a priori simulation (Exp. #1); (C1-C5) GEOS-Chem
a posteriori simulation by assimilating OMI $O_3$ observations (Exp. #8). (D1-D5) Bias in the a
priori simulations (Exp. #1 minus #8). (E1-E5) Effects of optimization on regional $O_3$
background conditions (Exp. #9 minus #8); (F1-F5) Effects of the usage of row-isolated data
(Exp. #10 minus #8). The output $O_3$ profiles are convolved with OMI averaging kernels.

**Fig. 8.** Trends of tropospheric $O_3$ columns in 2015-2020 (annual and seasonal averages) from
(A1-A5) OMI observations; (B1-B5) GEOS-Chem a priori simulation (Exp. #1); (C1-C5)
GEOS-Chem a posteriori simulation by assimilating OMI $O_3$ observations (Exp. #8). (D1-D5)
Effects of optimization on regional $O_3$ background conditions (Exp. #9 minus #8); (E1-E5)
Effects of the usage of row-isolated data (Exp. #10 minus #8). The output $O_3$ profiles are
convolved with OMI averaging kernels.

**Fig. 9.** Tropospheric $O_3$ columns in 2015-2020 (annual and seasonal averages) from (A1-A5)
GEOS-Chem a priori simulation (Exp. #1); (B1-B5) Assimilations of MEE surface $O_3$
observations (Exp. #5); (C1-C5) Assimilations of OMI $O_3$ observations (Exp. #8). (D1-D5)
Difference in tropospheric $O_3$ columns calculated by OMI-based assimilations minus MEE-
based assimilations (Exp. #8 minus #5). (E1-E5) Effects of seasonal variabilities in background
$O_3$ (Exp. #3 minus #1); (F1-F5) Effects of $O_3$ formation within the North China Plain PBL
(Exp. #1 minus #4). The output $O_3$ profiles are NOT convolved with OMI averaging kernels.

**Fig. 10.** Trends of tropospheric $O_3$ columns in 2015-2020 (annual and seasonal averages) from
(A1-A5) GEOS-Chem a priori simulation (Exp. #1); (B1-B5) Assimilations of MEE surface
$O_3$ observations (Exp. #5); (C1-C5) Assimilations of OMI $O_3$ observations (Exp. #8). (D1-D5)
Effects of interannual variabilities in background $O_3$ (Exp. #1 minus #2). The output $O_3$ profiles
are NOT convolved with OMI averaging kernels.

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

| | Experiments | Observations | O3 Boundary Conditions | Other Settings |
|---|---|---|---|---|
| A priori Simulations | #1 (Main) | N/A | Original (2015-2020) | |
| | #2 | N/A | Original (2015, fixed) | |
| | #3 | N/A | Original (2015-2020, fixed in spring) | |
| | #4 | N/A | Original (2015-2020) | PO3 = 0 (NCP) |
| Kalman Filter Assimilations | #5 (Main) | MEE | Original (2015-2020) | $\gamma = 0.8$ |
| | #6 | MEE | Original (2015-2020) | $\gamma = 0.0$ |
| | #7 | MEE | Original (2015-2020) | $\gamma = 1.0$ |
| | #8 (Main) | OMI | Optimized (2015-2020) | positions: 4-11 |
| | #9 | OMI | Original (2015-2020) | positions: 4-11 |
| | #10 | OMI | Optimized (2015-2020) | positions: 4-27 |

**Table. 1.** Single $O_3$ tracer simulation and assimilation experiments (Exp.) conducted in this work. Exp. #1: the main a priori simulation; Exp. #2: $O_3$ boundary conditions and stratospheric $O_3$ concentrations are fixed in 2015; Exp. #3: $O_3$ boundary conditions and stratospheric $O_3$ concentrations are fixed in the spring; Exp. #4: $O_3$ formation rates within the North China Plain PBL are set to zero; Exp. #5: the main assimilation by assimilating MEE surface $O_3$ observations with $\gamma = 0.8$; Exp. #6: only surface $O_3$ concentrations are adjusted ($\gamma = 0$); Exp. #7: full mixing of $O_3$ biases within the PBL ($\gamma = 1.0$); Exp. #8: the main assimilation by assimilating OMI $O_3$ observations; Exp. #9: $O_3$ boundary conditions are not optimized; Exp. #10: assimilating OMI $O_3$ observations at across-track positions 4-27.

| E. China (2015-2020) | | Annual | | Spring | | Summer | | Autumn | | Winter | |
|---|---|---|---|---|---|---|---|---|---|---|---|---|
| | | Mean | Trend | Mean | Trend | Mean | Trend | Mean | Trend | Mean | Trend |
| T2.1 surface (sampled) | MEE | 42.1±0.3 | 1.77±0.38 | 48.4±0.4 | 2.25±0.46 | 51.7±0.6 | 1.70±0.64 | 39.8±0.4 | 2.01±0.60 | 29.6±0.2 | 1.14±0.49 |
| | a priori | 43.2±0.2 | 0.21±0.13 | 48.0±0.2 | 0.31±0.15 | 56.3±0.5 | -0.12±0.38 | 40.1±0.3 | 0.45±0.19 | 28.5±0.3 | 0.40±0.17 |
| | KF-MEE | 41.8±0.2 | 1.24±0.28 | 47.2±0.3 | 1.60±0.34 | 51.7±0.5 | 1.16±0.55 | 39.5±0.3 | 1.47±0.47 | 29.5±0.2 | 0.80±0.37 |
| T2.2 surface | a priori | 42.6±0.1 | 0.10±0.11 | 47.7±0.1 | 0.16±0.11 | 53.1±0.2 | -0.19±0.29 | 39.1±0.1 | 0.25±0.19 | 30.8±0.2 | 0.35±0.13 |
| | KF-MEE | 41.3±0.1 | 0.55±0.17 | 46.7±0.1 | 0.71±0.17 | 49.8±0.2 | 0.36±0.36 | 38.0±0.1 | 0.69±0.31 | 31.0±0.2 | 0.54±0.19 |
| T2.3 trop. column (convolved) | OMI | 38.0±0.2 | -0.30±0.19 | 40.9±0.2 | 0.12±0.20 | 45.9±0.2 | -0.66±0.44 | 34.6±0.2 | -0.41±0.30 | 30.4±0.2 | -0.48±0.40 |
| | a priori | 37.1±0.1 | 0.02±0.14 | 41.0±0.2 | 0.17±0.24 | 43.2±0.2 | -0.19±0.16 | 32.6±0.1 | 0.15±0.19 | 31.3±0.2 | -0.06±0.18 |
| | KF-OMI | 37.9±0.1 | -0.17±0.15 | 41.1±0.2 | 0.08±0.07 | 45.5±0.2 | -0.51±0.37 | 34.2±0.1 | -0.17±0.24 | 30.7±0.1 | -0.17±0.23 |
| T2.4 trop. Column | a priori | 38.3±0.1 | 0.07±0.14 | 42.8±0.2 | -0.02±0.46 | 42.5±0.2 | 0.02±0.16 | 33.3±0.1 | 0.29±0.11 | 34.8±0.2 | 0.09±0.32 |
| | KF-MEE | 37.9±0.1 | 0.17±0.16 | 42.6±0.2 | 0.09±0.47 | 41.8±0.2 | 0.17±0.15 | 33.0±0.1 | 0.38±0.12 | 34.7±0.2 | 0.12±0.32 |
| | KF-OMI | 38.8±0.1 | -0.10±0.25 | 42.9±0.2 | -0.17±0.57 | 44.1±0.2 | -0.22±0.26 | 34.4±0.1 | 0.04±0.12 | 34.2±0.2 | -0.02±0.30 |

**Table. 2.** Averages (with units ppb or DU) and trends (with units ppb yr$^{-1}$ or DU yr$^{-1}$) of surface and tropospheric column O$_3$ concentrations in 2015-2020 over E. China from observations (MEE and OMI) and a priori (Exp. #1) and a posteriori (KF) simulations (Exp. #5 and #8). The domain definition of E. China is shown by Fig. 1A. T2.1): the modeled surface O$_3$ is sampled at the locations and times of MEE surface O$_3$ observations; T2.2): the modeled surface O$_3$ is averaged over E. China (land only); T2.3): the output O$_3$ profiles from the a priori and a posteriori simulations are convolved with OMI O$_3$ averaging kernels; T2.4): the output O$_3$ profiles are NOT convolved with OMI O$_3$ averaging kernels. The uncertainties in the averages are calculated using the bootstrapping method. The trends and uncertainties in the trends are calculated using the linear fitting of averages by using the least squares method (see details in the SI).

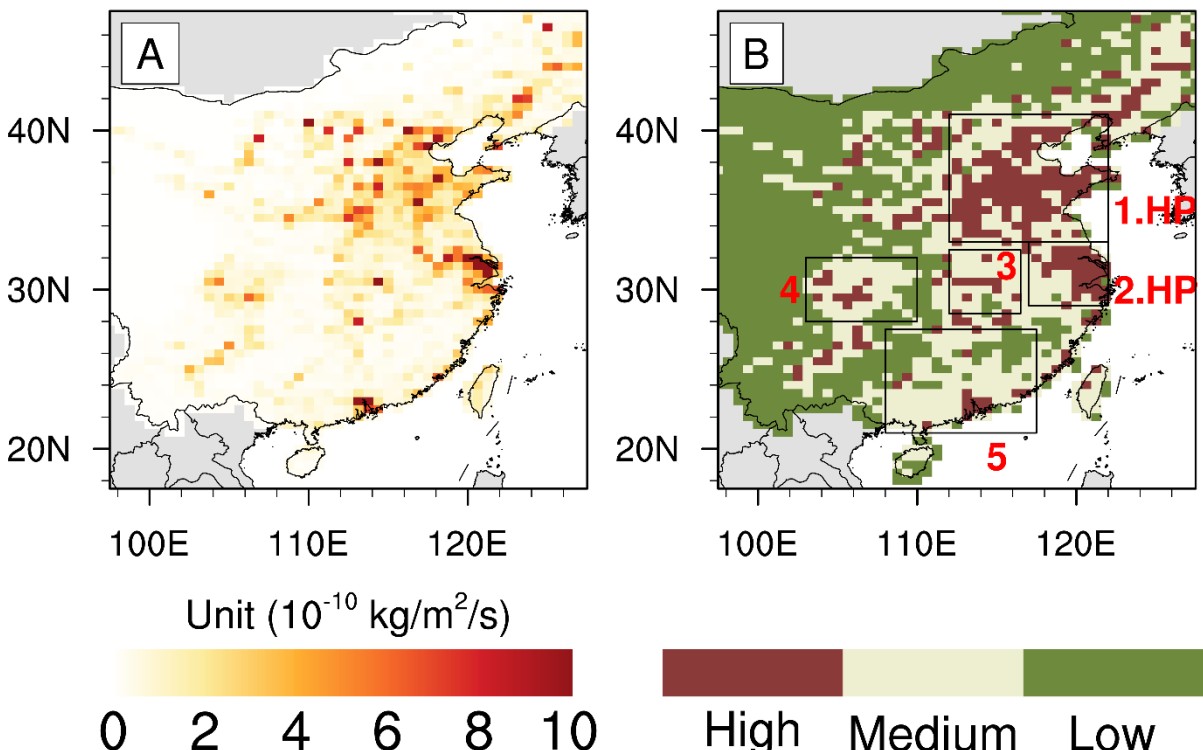

**Fig. 1.** (A) Anthropogenic $NO_x$ emissions over E. China in 2015; (B) Region definitions for the North China Plain (#1), Yangtze River Delta (#2), Central China (#3), Sichuan Basin (#4) and Southern China (#5). The different colors (red, gray and green) represent grids with high (highest 15%), medium (15-50%) and low (lowest 50%) anthropogenic $NO_x$ emissions. Regions #1 and #2 are defined as highly polluted (HP) regions by excluding grids with low and medium anthropogenic $NO_x$ emissions.

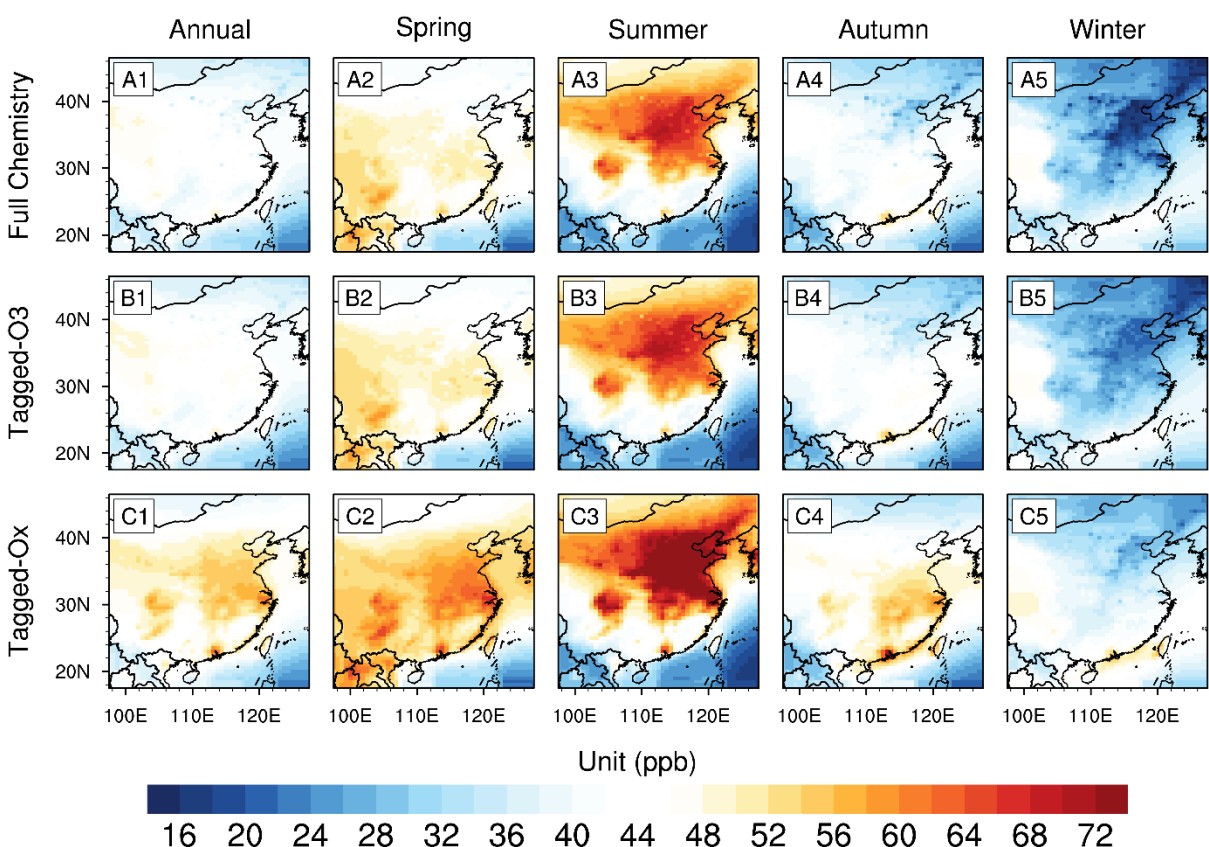

**Fig. 2.** Surface MDA8 $O_3$ in 2015-2020 (annual and seasonal averages) simulated by GEOS-Chem model with (A1-A5) full chemistry mode; (B1-B5) single $O_3$ tracer (tagged-$O_3$) mode; and (C1-C5) tagged-$O_x$ mode. The 8-hour range of surface $O_x$ is selected according to the time range of MDA8 $O_3$.

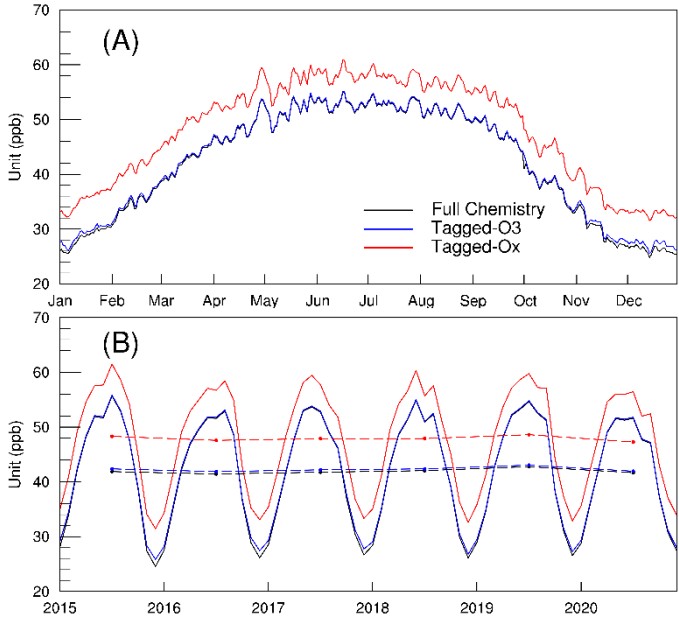

**Fig. 3.** (A) Daily averages of surface MDA8 $O_3$ over E. China in 2015-2020 from GEOS-Chem full chemistry (black), single $O_3$ tracer (tagged-$O_3$) (blue) and tagged-$O_x$ (red) simulations; (B) Monthly averages of MDA8 $O_3$. The dashed lines in panel B are annual averages.

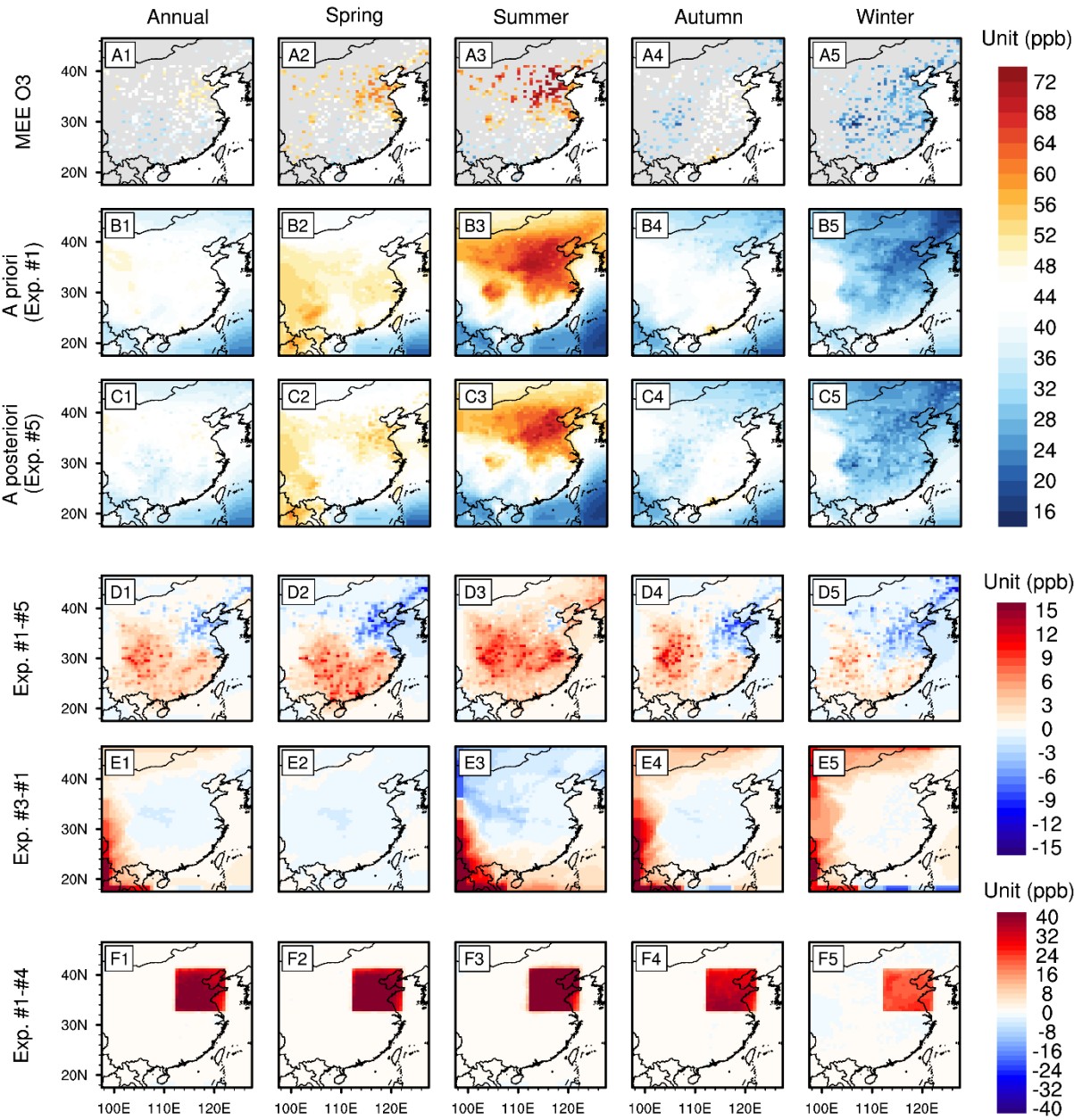

**Fig. 4.** Surface MDA8 O$_3$ in 2015-2020 (annual and seasonal averages) from (A1-A5) MEE stations; (B1-B5) GEOS-Chem a priori simulation (Exp. #1); (C1-C5) GEOS-Chem a posteriori simulation by assimilating MEE O$_3$ observations (Exp. #5); (D1-D5) Bias in the a priori simulations (Exp. #1 minus #5). (E1-E5) Effects of seasonal variabilities in background O$_3$ (Exp. #3 minus #1); (F1-F5) Effects of O$_3$ formation within the North China Plain PBL (Exp. #1 minus #4).

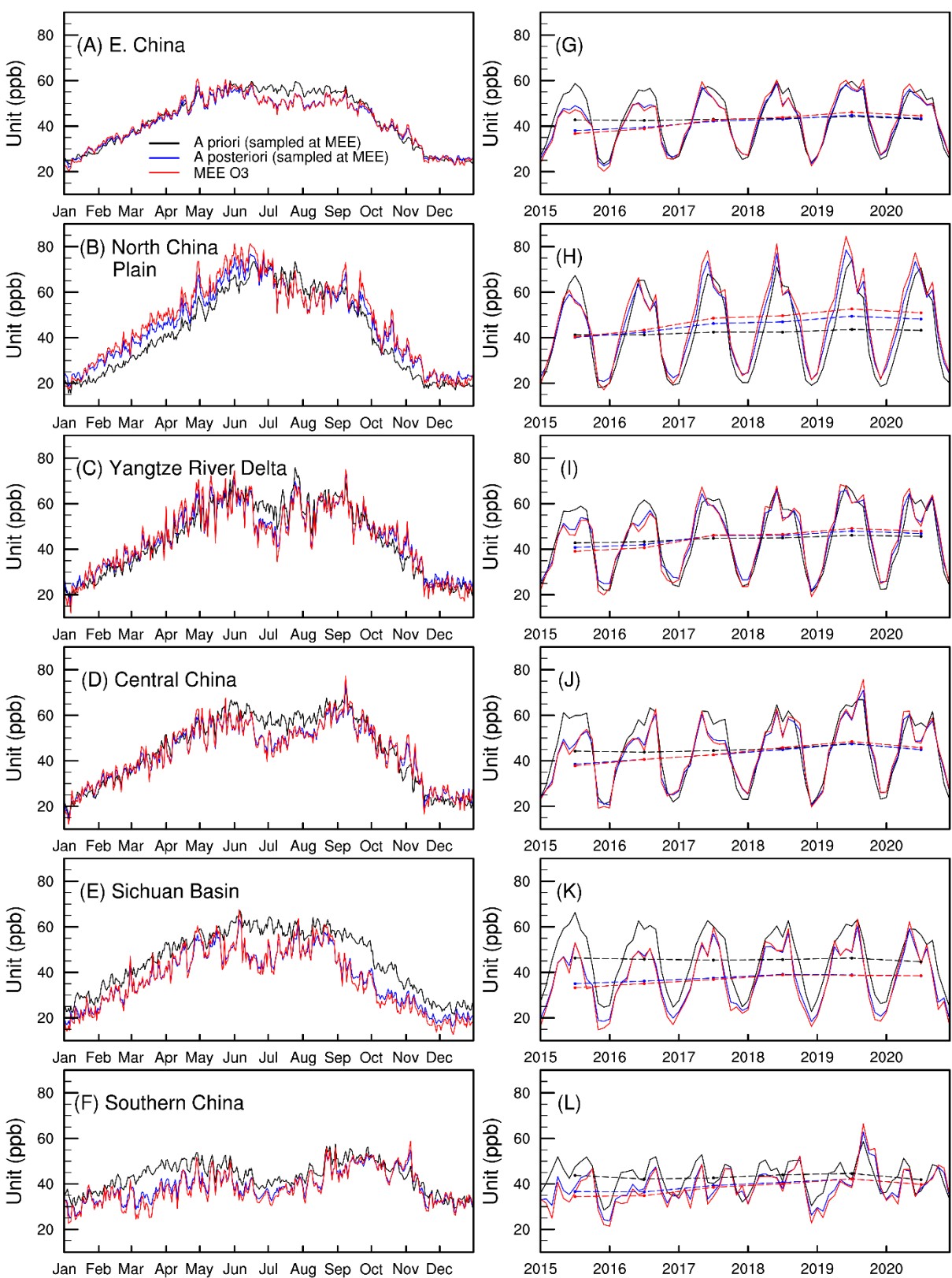

**Fig. 5.** (A-F) Daily averages of surface MDA8 $O_3$ in 2015-2020 from MEE stations (red) and GEOS-Chem a priori (black, Exp. #1) and a posteriori (blue, Exp. #5) simulations by assimilating MEE $O_3$ observations. (G-L) Monthly averages of MDA8 $O_3$. The dashed lines in panels G-L are annual averages. The domain definition of E. China is shown by Fig. 1A.

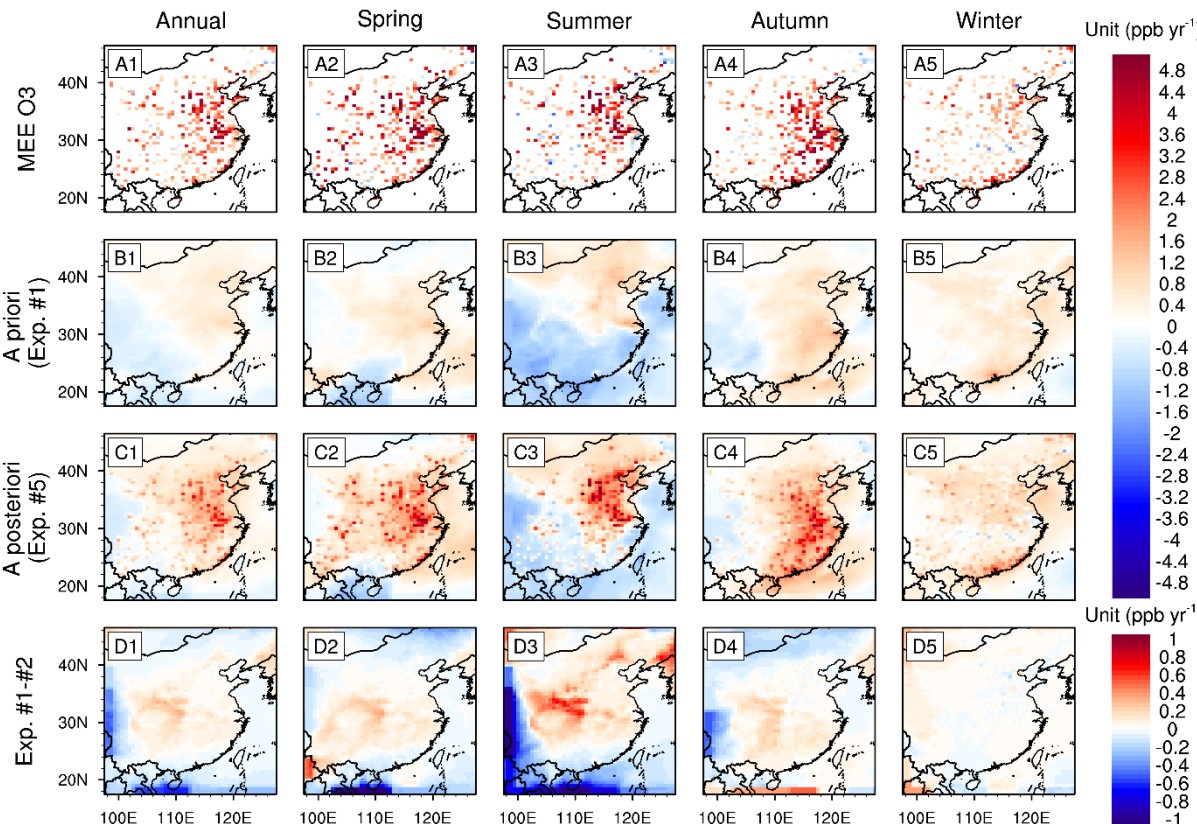

**Fig. 6.** Trends of surface MDA8 O$_3$ in 2015-2020 (annual and seasonal averages) from (A1-A5) MEE stations; (B1-B5) GEOS-Chem a priori simulation (Exp. #1); (C1-C5) GEOS-Chem a posteriori simulation by assimilating MEE O$_3$ observations (Exp. #5). (D1-D5) Effects of interannual variabilities in background O$_3$ (Exp. #1 minus #2).

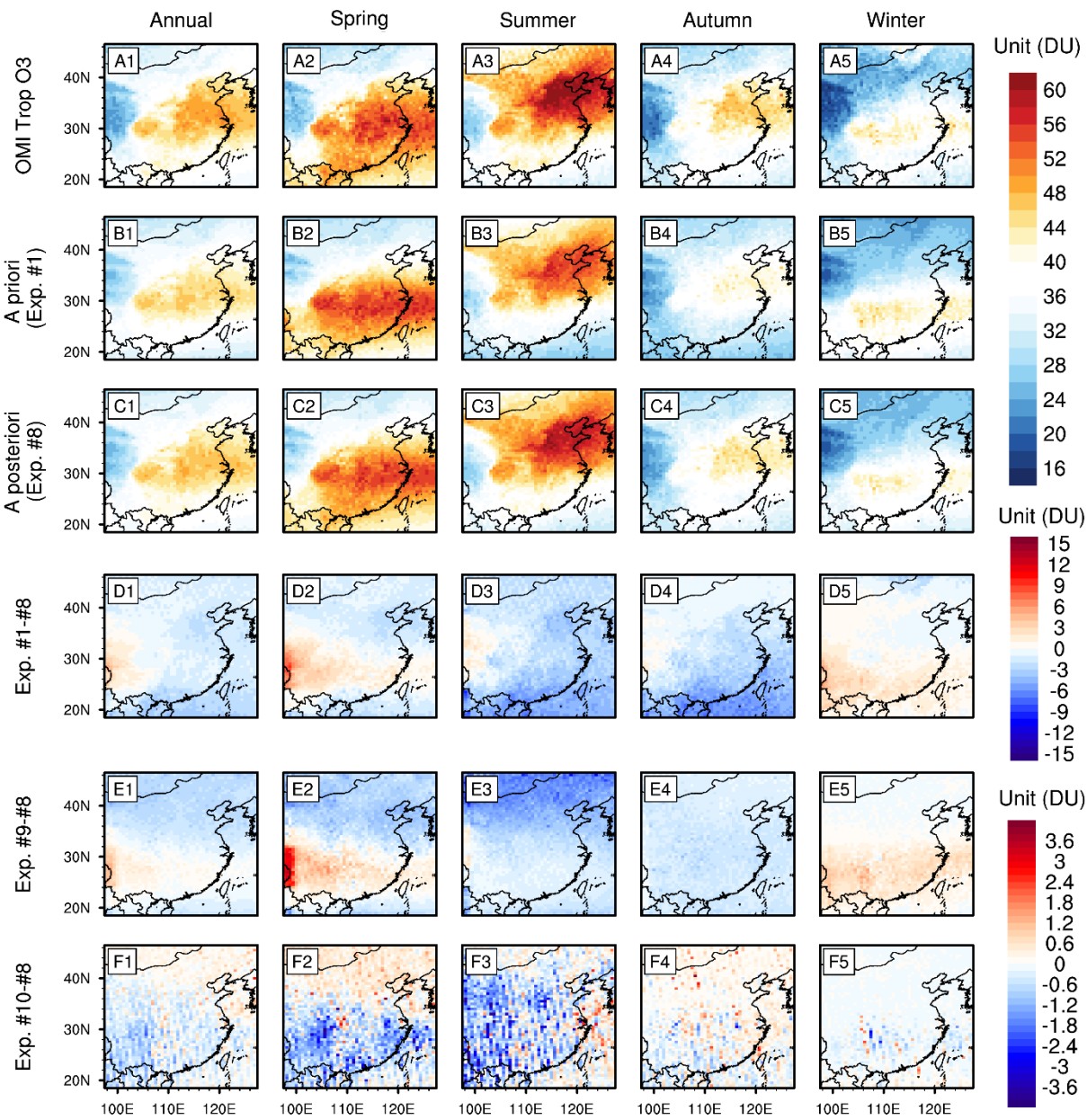

**Fig. 7.** Tropospheric $O_3$ columns in 2015-2020 (annual and seasonal averages) from (A1-A5) OMI observations; (B1-B5) GEOS-Chem a priori simulation (Exp. #1); (C1-C5) GEOS-Chem a posteriori simulation by assimilating OMI $O_3$ observations (Exp. #8). (D1-D5) Bias in the a priori simulations (Exp. #1 minus #8). (E1-E5) Effects of optimization on regional $O_3$ background conditions (Exp. #9 minus #8); (F1-F5) Effects of the usage of row-isolated data (Exp. #10 minus #8). The output $O_3$ profiles are convolved with OMI averaging kernels.

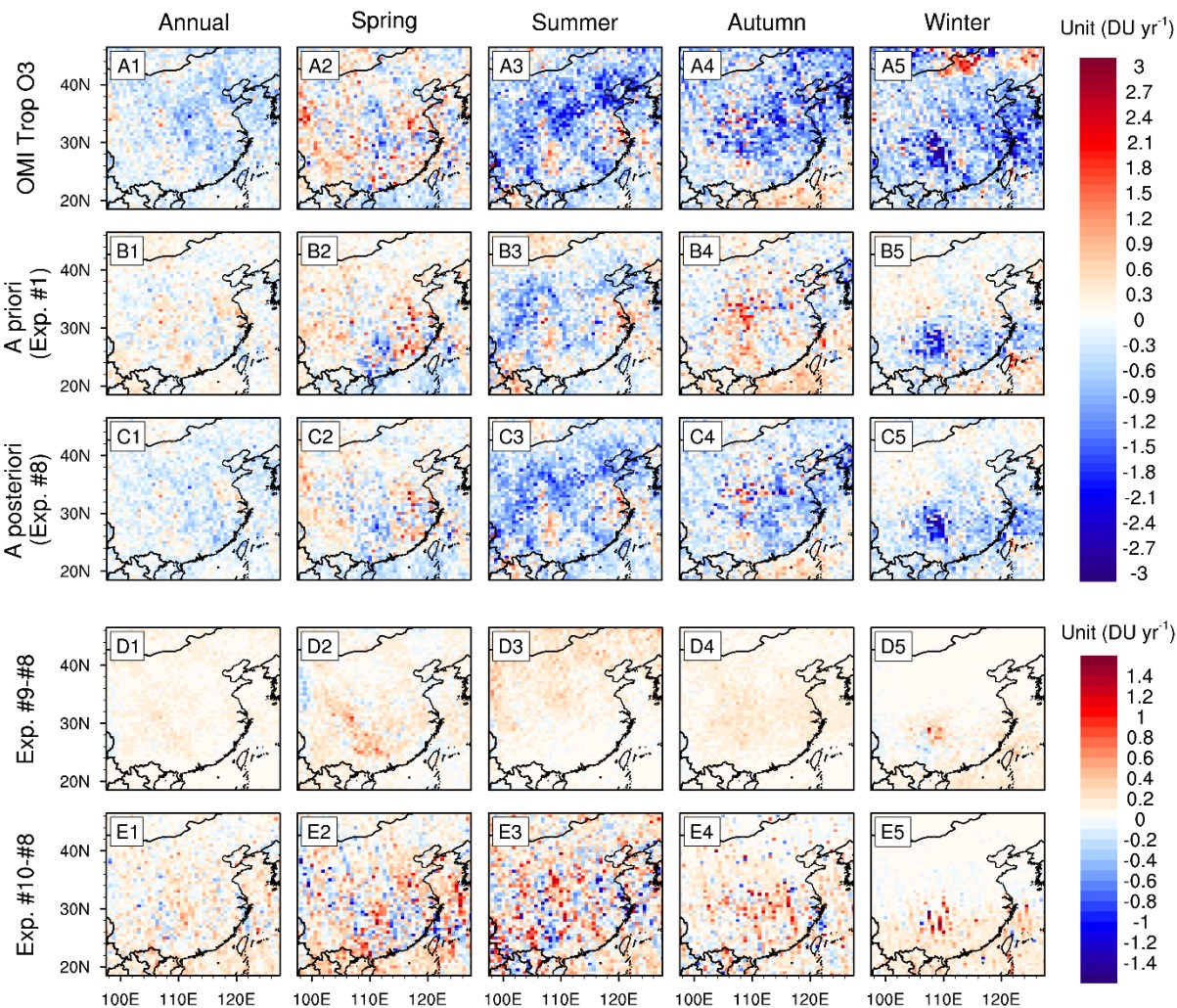

**Fig. 8.** Trends of tropospheric O₃ columns in 2015-2020 (annual and seasonal averages) from (A1-A5) OMI observations; (B1-B5) GEOS-Chem a priori simulation (Exp. #1); (C1-C5) GEOS-Chem a posteriori simulation by assimilating OMI O₃ observations (Exp. #8). (D1-D5) Effects of optimization on regional O₃ background conditions (Exp. #9 minus #8); (E1-E5) Effects of the usage of row-isolated data (Exp. #10 minus #8). The output O₃ profiles are convolved with OMI averaging kernels.

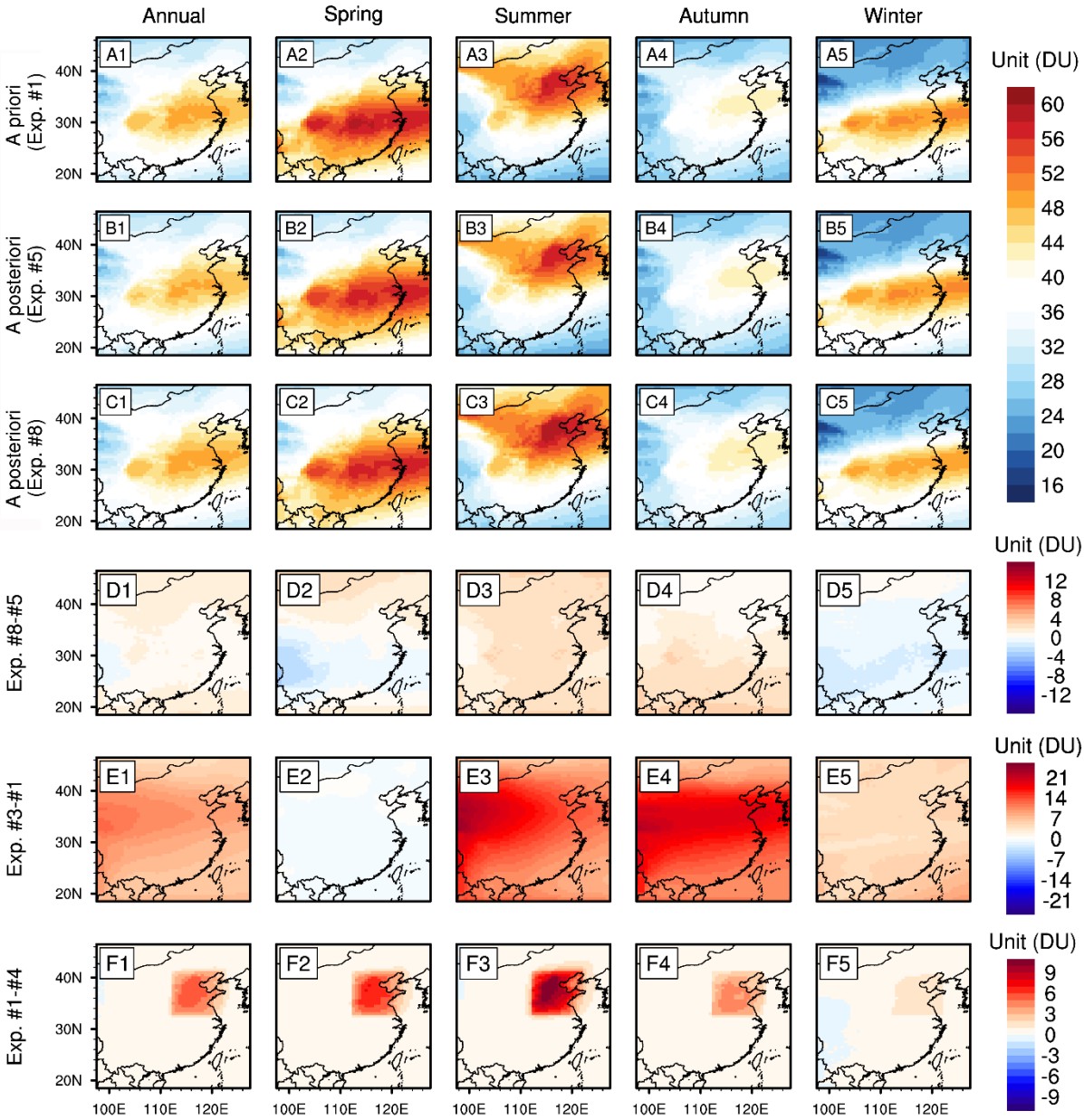

**Fig. 9.** Tropospheric O₃ columns in 2015-2020 (annual and seasonal averages) from (A1-A5) GEOS-Chem a priori simulation (Exp. #1); (B1-B5) Assimilations of MEE surface O₃ observations (Exp. #5); (C1-C5) Assimilations of OMI O₃ observations (Exp. #8). (D1-D5) Difference in tropospheric O₃ columns calculated by OMI-based assimilations minus MEE-based assimilations (Exp. #8 minus #5). (E1-E5) Effects of seasonal variabilities in background O₃ (Exp. #3 minus #1); (F1-F5) Effects of O₃ formation within the North China Plain PBL (Exp. #1 minus #4). The output O₃ profiles are NOT convolved with OMI averaging kernels.

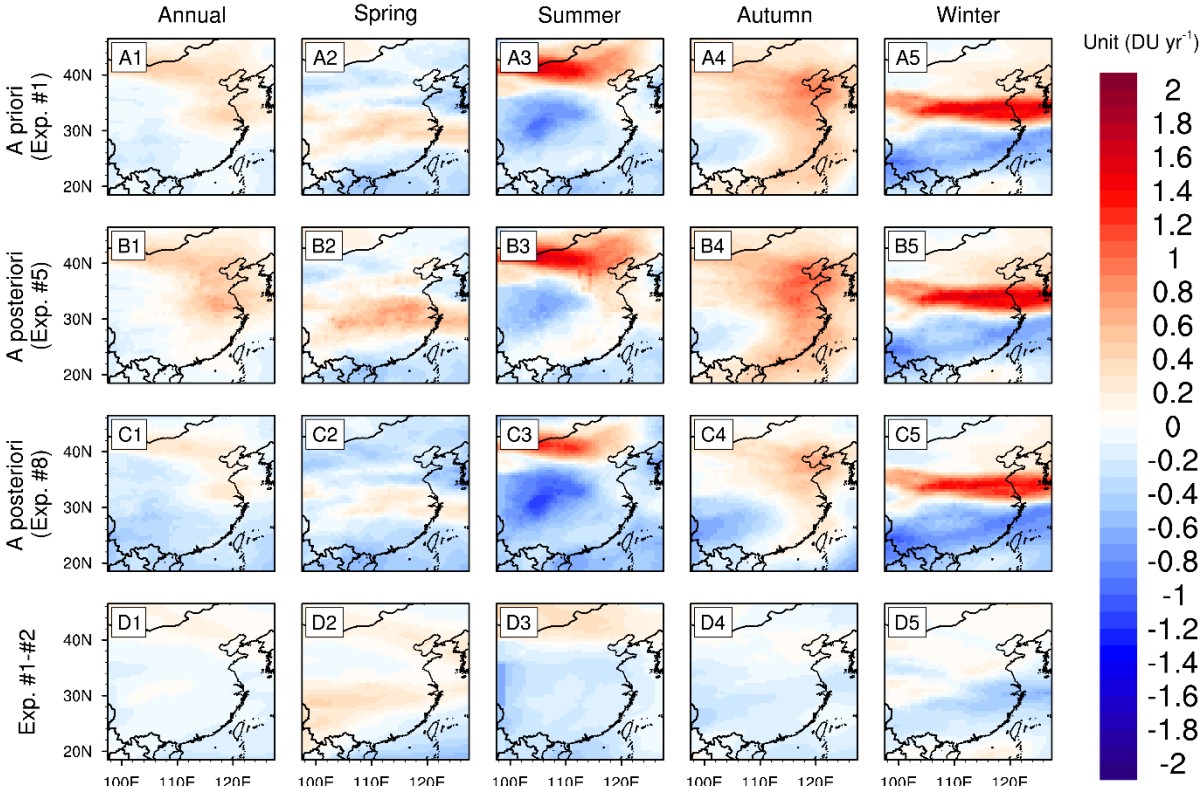

**Fig. 10.** Trends of tropospheric $O_3$ columns in 2015-2020 (annual and seasonal averages) from (A1-A5) GEOS-Chem a priori simulation (Exp. #1); (B1-B5) Assimilations of MEE surface $O_3$ observations (Exp. #5); (C1-C5) Assimilations of OMI $O_3$ observations (Exp. #8). (D1-D5) Effects of interannual variabilities in background $O_3$ (Exp. #1 minus #2). The output $O_3$ profiles are NOT convolved with OMI averaging kernels.