# Peer review of "Rapid O3 assimilations – Part 1: background and local contributions to tropospheric O3 changes in China in 2015-2020"

_Geoscientific Model Development, 2023_

## Author Response (AR1)

We thank the reviewers for their thoughtful and detailed comments. We have revised this manuscript carefully based on the comments. Below we respond to the individual comments.

Reviewer #1

**Question**: This paper describes the rapid assimilation of surface and column ozone into the GEOS-Chem model and uses this approach to quantify changes in ozone over China between 2015 and 2020. It provides useful confirmation of the ozone changes over this important and rapidly-evolving region. However, the paper has a number of major deficiencies that make it unsuitable for publication in GMD in its present form. Specifically, the purpose and rationale of the study are not stated; the motivation for a single-tracer assimilation is not described; the benefits of "rapid" assimilation are not explained; and the results for ozone trends are not compared with those from any previous studies to provide context for the reader. These are major deficiencies that need to be addressed fully before the paper can be considered for publication.

**Answer**: We appreciate the reviewer for the constructive comments which have helped us largely in improving our paper. We have made extensive revisions to the paper, particularly, the experiments have been extended from three (Exp. #1, #5 and #8) to 10 (see the new Table 1). Beyond the original paper, these new sensitivity experiments allow us to provide more mechanism analysis about the impacts of background $O_3$ (particularly, the interannual and seasonal variabilities in the background $O_3$ as well as optimization in the background $O_3$) and local $O_3$ formation on the changes of surface and free tropospheric $O_3$ over E. China. The title of this manuscript has been changed to "Rapid $O_3$ assimilations – Part 1: background and local contributions to tropospheric $O_3$ changes in China in 2015-2020" to match the extended scientific discussions.

Furthermore, it should be noted that only 270 hours (wall time) are required to finish the 10 experiments with the single $O_3$ tracer mode, which is 94% lower than the full chemistry simulations (4812 hours, wall time). It demonstrates the benefits of the single $O_3$ tracer mode.

**Question**: I have looked at Part 2 of this study under consideration for ACP (also a weak paper) and would suggest that the two papers are combined into a single paper. It makes little scientific sense to split the study into two given that the tools, approach and aims are the same, and only the regions of interest differ. This would address a number of obvious problems, and the combined paper would be more complete, making a more useful addition to the published literature.

**Answer**: In our recent study (Jiang et al. 2022), we found a slowdown of declines in tropospheric $NO_2$ columns with respect to surface $NO_2$ concentrations over both the US and Europe since 2010, which is partially attributed to the enhanced contribution from free tropospheric $NO_2$ background. The objective of the Part 2 manuscript (in ACP) is to evaluate the possible effects of different trends in surface and free tropospheric $NO_2$ on surface and free tropospheric $O_3$ over the US and Europe in 2005-2020. The Part 2 manuscript has been revised with a new title "Rapid $O_3$ assimilations – Part 2:

tropospheric $O_3$ changes accompanied by declines in $NO_x$ emissions in the US and Europe in 2005-2020" to emphasize this objective.

However, Jiang et al. (2022) find broadly consistent trends in surface and free tropospheric $NO_2$ over China. It is thus difficult to combine these two papers because the targeted objective of the Part 2 manuscript is not valid for the $NO_2$ and $O_3$ issues in China. Consequently, these two studies were organized as individual papers initially.

We acknowledge that the original Part 1 manuscript (in GMD) is weak, and hope the revised manuscript with extended scientific discussions by including 10 experiments can provide more useful information for $O_3$ changes in China.

References:

Jiang, Z., Zhu, R., Miyazaki, K., McDonald, B. C., Klimont, Z., Zheng, B., Boersma, K. F., Zhang, Q., Worden, H., Worden, J. R., Henze, D. K., Jones, D. B. A., Denier van der Gon, H. A. C., and Eskes, H.: Decadal Variabilities in Tropospheric Nitrogen Oxides Over United States, Europe, and China, J Geophys Res-Atmos, 127, e2021JD035872, 10.1029/2021jd035872, 2022.

General Comments

**Question**: The term "tagged-O3 mode" is not explained here, and is confusing as no tagging is used anywhere in this study. Please explain what this term means in the present context. "Single tracer mode" would be a much clearer and more appropriate description given that the model is run with a single ozone tracer.

**Answer**: As the reviewer suggested, the usage of the term "tagged-$O_3$" was reduced and was substituted with "single $O_3$ tracer". The term "tagged-$O_3$" was only mentioned three times in the revised version in the Introduction, Methodology and Conclusion Sections to facilitate the readers who are familiar with the GEOS-Chem model.

**Question**: What is the purpose of the ozone assimilation? The apparent goal of the study (to identify ozone changes in China) can be performed from the observations alone, so what additional information does assilimation provide? This needs to be explained clearly. The introduction does not provide a rationale for the approach or explain why it is needed.

**Answer**: As clarified in the revised manuscript: "Data assimilations, by combining modeled and observed $O_3$ concentrations, can take advantage of both simulations and observations to produce more accurate $O_3$ concentrations". Furthermore, "Satellite instruments provide globally covered $O_3$ observations that are sensitive to $O_3$ concentrations in the free troposphere. The OMI-based assimilations can thus reflect the optimized adjustments in both global background and local $O_3$ concentrations. On the other hand, surface observations are sensitive to local $O_3$ concentrations. Surface observation-based assimilations can reflect the optimized adjustments in local contributions, and the information of local contributions can be transported into the free troposphere via vertical convection in the assimilation processes, which is different from the fusion of satellite and surface observations".

Consequently, the purpose of this work is to provide a comparative analysis by assimilating satellite and surface $O_3$ observations to provide a better characterization of

O$_3$ changes in the surface and free troposphere.

**Question**: What are the benefits of "rapid" assimilation? The analysis described in the paper could be performed equally well with assimilation using the full model, so what value does the speed up provide? I can see a benefit if the assimilation is to be repeated a very large number of times, but this does not appear to be the case here.

**Answer**: The experiments in the revised manuscript have been extended from three to 10. The usage of the single O$_3$ tracer mode leads to a dramatic reduction in the computation cost from 4812 to 270 hours (wall time). It demonstrates the benefits of the single O$_3$ tracer mode.

**Question**: How do the ozone trends over China compare with previous observational or model-based studies? Many recent studies have quantified these, so it is essential to provide a comparison as context for the reader and to demonstrate the value of the approach adopted here.

**Answer**: The comparison between this work and other studies was added in the revised version, for example: "The annual increasing trend (1.24 ppb yr$^{-1}$) in the assimilated surface O$_3$ concentrations is more consistent with MEE O$_3$ observations (1.77 ppb yr$^{-1}$) which are comparable with the reported recent trends in surface O$_3$ concentrations in China of 1.25-2.0 ppb yr$^{-1}$ (Mousavinezhad et al., 2021; Wei et al., 2022; Wang, W. et al., 2022)".

References:

Mousavinezhad, S., Choi, Y., Pouyaei, A., Ghahremanloo, M., and Nelson, D. L.: A comprehensive investigation of surface ozone pollution in China, 2015–2019: Separating the contributions from meteorology and precursor emissions, Atmospheric Research, 257, 10.1016/j.atmosres.2021.105599, 2021.

Wang, W., Parrish, D. D., Wang, S., Bao, F., Ni, R., Li, X., Yang, S., Wang, H., Cheng, Y., and Su, H.: Long-term trend of ozone pollution in China during 2014–2020: distinct seasonal and spatial characteristics and ozone sensitivity, Atmos Chem Phys, 22, 8935-8949, 10.5194/acp-22-8935-2022, 2022.

Wei, J., Li, Z., Li, K., Dickerson, R. R., Pinker, R. T., Wang, J., Liu, X., Sun, L., Xue, W., and Cribb, M.: Full-coverage mapping and spatiotemporal variations of ground-level ozone (O3) pollution from 2013 to 2020 across China, Remote Sens Environ, 270, 10.1016/j.rse.2021.112775, 2022.

**Question**: There is no consideration of uncertainty in the results, or any attempt to explore the significance of the derived trends. While the a posteriori results are compared with the a priori model results, no attempt is made to investigate or explain why the a priori results might be wrong, and this is a missed opportunity.

**Answer**: Uncertainty estimates have been included in all Tables in the revised manuscript. As described in the revised manuscript: "The uncertainties in the averages are calculated using the bootstrapping method. The trends and uncertainties in the trends are calculated using the linear fitting of averages by using the least squares method (see details in the SI)". Thank the reviewer for pointing out this issue!

Furthermore, the single O$_3$ tracer simulation is driven by the archived production and loss of O$_3$ provided by the full chemistry simulation. It is a useful tool to analyze the sources and transport of tropospheric O$_3$, for example, Zhang et al. (2008), Zhu et al. (2017) and Han et al. (2018) by using the tagged-O$_x$ mode as well as the extended analysis of background and local contributions in this work. However, it may not be an ideal choice to explain the inaccurate simulation of O$_3$ photochemistry.

References:

Zhang, L., Jacob, D. J., Boersma, K. F., Jaffe, D. A., Olson, J. R., Bowman, K. W., Worden, J. R., Thompson, A. M., Avery, M. A., Cohen, R. C., Dibb, J. E., Flock, F. M., Fuelberg, H. E., Huey, L. G., McMillan, W. W., Singh, H. B., and Weinheimer, A. J.: Transpacific transport of ozone pollution and the effect of recent Asian emission increases on air quality in North America: an integrated analysis using satellite, aircraft, ozonesonde, and surface observations, Atmos Chem Phys, 8, 6117-6136, DOI 10.5194/acp-8-6117-2008, 2008.

Zhu, Y., Liu, J., Wang, T., Zhuang, B., Han, H., Wang, H., Chang, Y., and Ding, K.: The Impacts of Meteorology on the Seasonal and Interannual Variabilities of Ozone Transport From North America to East Asia, J Geophys Res-Atmos, 122, 10,612-610,636, 10.1002/2017jd026761, 2017.

Han, H., Liu, J., Yuan, H., Zhuang, B., Zhu, Y., Wu, Y., Yan, Y., and Ding, A.: Characteristics of intercontinental transport of tropospheric ozone from Africa to Asia, Atmos Chem Phys, 18, 4251-4276, 10.5194/acp-18-4251-2018, 2018.

**Question**: There is an over-reliance on referencing recent papers for general material such as that presented at the start of the introduction. This suggests that the authors are not familiar with the wider literature. Where references are needed, please cite original primary references and not recent derivative studies.

**Answer**: More original primary references were cited in the revised manuscript.

Specific Comments:

**Question**: Line 44: the emphasis of this sentence reveals some weaknesses in perspective. Surface stations provide the most direct information on air quality, while satellite observations are much less important for this. Line 46: "assimilation of surface observations can effectively improve the predicted surface O3 concentrations". This statement appears obvious; please explain why it is useful.

**Answer**: These sentences have been revised: "The important role of O$_3$ in the atmosphere has led to many efforts focusing on O$_3$ observations that have improved our understanding of atmospheric O$_3$ ( Logan et al., 2012; Oetjen et al., 2016; Parrish et al., 2021). The limited spatial coverage of O$_3$ observations promotes the efforts of spatial extensions of O$_3$ observations ( Chang et al., 2015; Peng et al., 2016). Recent advances in machine learning techniques further provide a new method to extend O$_3$ observations by fusing satellite and surface observations (Li et al., 2020; Liu et al., 2022; Wei et al., 2022)".

**Question**: Line 52: better understanding requires photochemistry, not assimilation; this sentence is incorrect and needs rephrasing.

**Answer**: Thank the reviewer for pointing out this issue! The challenge of accurate simulation of photochemistry was emphasized in the revised version: "Chemical transport models (CTMs), as powerful tools, have been widely used to simulate and interpret observed $O_3$ variabilities (Parrington et al., 2012; Jiang et al., 2016; Li et al., 2019). Despite the advances in CTMs, an accurate simulation of observed $O_3$ is still challenging because of uncertainties in physical and chemical processes (Peng et al., 2021; Chen et al., 2022), emission inventories (Elguindi et al., 2020; Jiang et al., 2022 and coarse model resolutions (Schaap et al., 2015; Benavides et al., 2021). Furthermore, the high computational cost is a bottleneck for rapid simulations, which poses a possible barrier to better understanding tropospheric $O_3$".

**Question**: Lines 96-100: The AQS and AirBase data are not used in this paper, so this information should be removed unless the two parts of this study are combined. It would be more useful to comment on the locations of the MEE sites, in particular on how many are urban and suburban, and if any rural sites are available. What issues are associated with using a measurement network that is predominantly urban?

**Answer**: As discussed in the revised manuscript: "These real-time monitoring stations report hourly concentrations of criteria pollutants from 1691 sites in 2020. All stations (1441 urban sites and 250 urban background sites) are assimilated in our analysis … It should be noted that the assimilation of $O_3$ observations from urban and urban background sites may result in possible overestimation of surface $O_3$ concentrations over the rural areas".

Furthermore, as the reviewer suggested, the description of the AQS and AirBase data has been removed in the revision.

**Question**: Line 121-124: How does this convolution approach differ from previous studies? If this is standard, please cite the original literature.

**Answer**: The convolution is the standard treatment of OMI data. Two references were added in the revised manuscript:

Liu, X., Bhartia, P. K., Chance, K., Spurr, R. J. D., and Kurosu, T. P.: Ozone profile retrievals from the Ozone Monitoring Instrument, Atmos Chem Phys, 10, 2521-2537, 10.5194/acp-10-2521-2010, 2010.

Huang, G., Liu, X., Chance, K., Yang, K., Bhartia, P. K., Cai, Z., Allaart, M., Ancellet, G., Calpini, B., Coetzee, G. J. R., Cuevas-Agulló, E., Cupeiro, M., De Backer, H., Dubey, M. K., Fuelberg, H. E., Fujiwara, M., Godin-Beekmann, S., Hall, T. J., Johnson, B., Joseph, E., Kivi, R., Kois, B., Komala, N., König-Langlo, G., Laneve, G., Leblanc, T., Marchand, M., Minschwaner, K. R., Morris, G., Newchurch, M. J., Ogino, S.-Y., Ohkawara, N., Piters, A. J. M., Posny, F., Querel, R., Scheele, R., Schmidlin, F. J., Schnell, R. C., Schrems, O., Selkirk, H., Shiotani, M., Skrivánková, P., Stübi, R., Taha, G., Tarasick, D. W., Thompson, A. M., Thouret, V., Tully, M. B., Van Malderen, R., Vömel, H., von der Gathen, P., Witte, J. C., and Yela, M.: Validation of 10-year SAO OMI Ozone Profile (PROFOZ) product using ozonesonde observations, Atmos Meas Tech, 10, 2455-2475, 10.5194/amt-10-2455-2017, 2017.

**Question**: Line 138 states that MEIC emissions are used, while Line 143 indicates that emissions are scaled corresponding to MEIC. Which of these approaches is used?

**Answer**: GEOS-Chem model (v12-8-1) uses MEIC/MIX inventory for anthropogenic emissions in the China/Asia domain. The reference year for MEIC/MIX inventory is 2010 with annual scaling factors in 2008-2010 in the GEOS-Chem model. Consequently, the annual total emissions must be adjusted manually in the simulations using additional annual scaling factors.

The description has been revised to make this point clearer: "Following Jiang et al. (2022), the total anthropogenic $NO_x$ and VOC emissions in the GEOS-Chem model are scaled based on Zheng et al. (2018) and Li, M. et al. (2019)". We are sorry for this confusion!

References:

Li, M., Zhang, Q., Zheng, B., Tong, D., Lei, Y., Liu, F., Hong, C., Kang, S., Yan, L., Zhang, Y., Bo, Y., Su, H., Cheng, Y., and He, K.: Persistent growth of anthropogenic non-methane volatile organic compound (NMVOC) emissions in China during 1990–2017: drivers, speciation and ozone formation potential, Atmos Chem Phys, 19, 8897-8913, 10.5194/acp-19-8897-2019, 2019.

Zheng, B., Tong, D., Li, M., Liu, F., Hong, C., Geng, G., Li, H., Li, X., Peng, L., Qi, J., Yan, L., Zhang, Y., Zhao, H., Zheng, Y., He, K., and Zhang, Q.: Trends in China's anthropogenic emissions since 2010 as the consequence of clean air actions, Atmos Chem Phys, 18, 14095-14111, 10.5194/acp-18-14095-2018, 2018.

**Question**: Line 148: information on US and Europe is provided here, but are not needed in this paper unless the papers are combined.

**Answer**: The information about the US and Europe has been removed.

**Question**: Line 168: "The model errors are assumed to be 50%." Why? Some explanation is needed here.

**Answer**: The manuscript has been revised: "The model errors are assumed to be 50% because the objective of our assimilations is to provide dynamic extensions of atmospheric $O_3$ observations. The a posteriori $O_3$ concentrations with the assumption of 50% model errors are expected to match better with atmospheric $O_3$ observations".

**Question**: Line 174: If the errors are calculated on a station basis, how is the grid-based superobservation applied? More information is required here.

**Answer**: More information of the superobservation method was added in the revised version:

"Furthermore, the "superobservation" method was applied in this work to further reduce the influence of representative error (Miyazaki et al., 2017; Tang et al., 2022):

$$\omega_j = 1/\varepsilon_j^2 \quad \text{(Eq. 5)}$$

$$y_s = \sum_{j=1}^{k} \omega_j y_j / \sum_{j=1}^{k} \omega_j \quad \text{(Eq. 6)}$$

$$1/\varepsilon_s^2 = \sum_{j=1}^{k} 1/\varepsilon_j^2 \quad \text{(Eq. 7)}$$

where $y_j$ is $O_3$ observation of the $j$th station, $\omega_j$ represents the weighting factor of the $j$th station, $y_s$ and $\varepsilon_s$ are the grid-based $O_3$ observations and errors (superobservation), respectively".

References:

Miyazaki, K., Eskes, H., Sudo, K., Boersma, K. F., Bowman, K., and Kanaya, Y.: Decadal changes in global surface $NO_x$ emissions from multi-constituent satellite data assimilation, Atmos Chem Phys, 17, 807-837, 10.5194/acp-17-807-2017, 2017.

Tang, Z., Chen, J., and Jiang, Z.: Discrepancy in assimilated atmospheric CO over East Asia in 2015–2020 by assimilating satellite and surface CO measurements, Atmos Chem Phys, 22, 7815-7826, 10.5194/acp-22-7815-2022, 2022.

**Question**: Line 201: Why is it necesary to "design and perform different assimilation experiments"? Is this just to improve the method? This is an important point, as the rationale for development of a rapid assimilation method depends upon it.

**Answer**: The new sensitivity experiments in the revised manuscript provide more mechanism analysis about the impacts of background $O_3$ and local $O_3$ formation on the changes of surface and free tropospheric $O_3$ over E. China. It demonstrates the necessity to design and perform different experiments.

**Question**: Line 211: The difference between tagged-O3 and tagged-Ox needs to be explained clearly to the reader before this (no one outside the GEOS-Chem community is likely to understand the distinction).

**Answer**: The discussion has been revised: "In contrast, the tagged-$O_x$ mode of the GEOS-Chem model is driven by the archived production and loss of $O_x$, which is the combination of multiple species ($O_x=O_3+NO_2+2NO_3+3N_2O_5+HNO_3+HNO_4$ +peroxyacylnitrates). There are large discrepancies between full chemistry (Fig. 2A1-A5) and tagged-$O_x$ (Fig. 2C1-C5) simulations". Thank the reviewer for this suggestion!

**Question**: Line 221-224: This point is poorly explained. The issue is associated with the timescales for turblent transport in the PBL vs. chemical timescales, and this justifies the need to adjust ozone throughout the PBL. Please explain how the factor of 0.8 was chosen.

**Answer**: Besides the original assimilation experiment ($\gamma = 0.8$ by assuming partial mixing of $O_3$ biases within the PBL), two new sensitivity experiments were added in the revised manuscript: "1) $\gamma = 0$ by assuming the biased surface $O_3$ concentrations are completely caused by biased $O_3$ production and loss at the surface level; 2) $\gamma = 1$ by assuming full mixing of $O_3$ biases within the PBL".

As discussed in the revised manuscript: "As shown in Fig. S2A (see the SI), the assimilated surface MDA8 $O_3$ concentrations show good agreement by using different $\gamma$ parameters: 42.3, 41.8 and 42.0 ppb ($\gamma = 0$, 0.8 and 1.0) in 2015-2020; there are noticeable discrepancies in the trends of assimilated surface $O_3$ concentrations: 0.80, 1.24 and 1.50 ppb yr$^{-1}$ ($\gamma = 0$, 0.8 and 1.0) in 2015-2020 (Fig. S2B), and the trends obtained by considering the mixing of $O_3$ biases ($\gamma = 0.8$ and 1.0) match better with MEE $O_3$ observations (1.77 ppb yr$^{-1}$). Fig. S3 (see the SI) further demonstrates

tropospheric $O_3$ columns by assimilating MEE $O_3$ observations in 2015-2020. We find good agreement in the assimilated tropospheric $O_3$ columns by using different $\gamma$ parameters, i.e., the mean tropospheric $O_3$ columns are 38.1, 37.9 and 37.9 DU, and the trends of tropospheric $O_3$ columns are 0.11, 0.17 and 0.21 ppb yr$^{-1}$ ($\gamma = 0$, 0.8 and 1.0). Considering the better agreement in the trends of assimilated surface $O_3$ concentrations ($\gamma = 0.8$ and 1.0) with observations, we finally decide to set $\gamma = 0.8$ as our main assimilation setting by assuming partial mixing of $O_3$ biases within the PBL".

**Question**: Para 332: This explanation is unconvincing, and clearer justification is required. The differing seasonality of the column over the NCP is interesting, but no evidence of transport differences is provided. It would be possible to diagnose this properly using the tagged-O3 approach for the purpose it was designed for.

**Answer**: Thank the reviewer for this suggestion! Two new sensitivity experiments were added to investigate the causes of the different seasonality between highly polluted North China Plain and other lower polluted regions. As discussed in the revised manuscript:

"The assimilated tropospheric $O_3$ columns are maximum in June-July over the highly polluted North China Plain and March-May over other lower polluted regions (Fig. S5, see the SI). Fig. 9E1-E5 exhibit the effects of seasonal variabilities in background $O_3$ (Exp. #3). The fixed background $O_3$ in the spring can result in dramatic increases in tropospheric $O_3$ columns by 14.3 (summer), 15.1 (autumn) and 4.8 (winter) DU over E. China. Fig. 9F1-F5 further exhibit the effects of $O_3$ formation within the North China Plain PBL (Exp. #4) on tropospheric $O_3$ columns, which are 5.4 (spring), 8.1 (summer), 3.6 (autumn) and 1.3 (winter) DU over the North China Plain. In addition, as shown in Fig. S6 (see the SI), there is a larger enhancement in $O_3$ production rates in the free troposphere (600-300 hPa) over the North China Plain in the summer than in other lower polluted regions. Consequently, the spring maximum in tropospheric $O_3$ columns over lower polluted regions is caused by the enhanced background $O_3$ (Fig. 9E1-E5), and the summer maximum in tropospheric $O_3$ columns over the highly polluted North China Plain is caused by the local contributions from enhanced $O_3$ formation within the North China Plain PBL (Fig. 9F1-F5) and free troposphere (Fig. S6)".

Typos and minor issues

**Question**: Line 91: "have the ability to report" Please rephrase this.

**Answer**: Changed: "These real-time monitoring stations report hourly concentrations".

**Question**: Line 129: "does not cancel out" - this point is poorly described, please rephrase.

**Answer**: Changed: "The unit for averaging kernels in this OMI product is DU/DU because the conversion from DU to ppb varies with altitude".

**Question**: Line 226: The "l" and "1" are too similar, please change the notation here (use "n" for the layer?)

**Answer**: Changed.

**Question**: Fig S1 is very unclear. What does this framework show, and what information is passed following the arrows? Which aspects are full chemistry and which are single tracer? This diagram needs to be reconsidered and redrawn.

**Answer**: This diagram has been redrawn. Thank the reviewer for pointing out this issue!

**Question**: A large number of multi-panel figures are included in the paper, but many of the results are only very briefly mentioned in the text. The paper would be sharper and clearer if the authors were more selective and moved some of these to the supplement.

**Answer**: The original Fig. 10 and Fig. 11 were moved to the supplement.

Reviewer #2

**Question**: In this study, the authors developed single tracer tagged-O3 mode of the GEOS-Chem model to investigate the tropospheric and surface NO2- and O3 changes in China. Further data assimilations were performed with both surface and satellite observation. The authors also pointed out a companion paper on ACPD which applied the method developed here to study the ozone changes in US and Europe.

Unfortunately, these two papers are not in companion order, either in ACP or GMD. However, reading from this manuscript, it was still drafted to be the case. I suggest the authors spend some time to reorganize the manuscript, so it will be independent of the other one. For example, in line 96-100: I suggest the authors leave these few sentences to the companion paper.

**Answer**: Thank the reviewer for the comments! The manuscript has been revised to enhance independence. The sentences in lines 96-100 have been removed.

**Question**: Line 21: give full name of E. Asia, and also E. China in line 25 since they appear for the first time. Pay attention to other abbreviations in the main context.

**Answer**: Thank the reviewer for this suggestion! The abbreviation has been checked and clarified.

---

## Author Response (AR2)

We thank the reviewer for the thoughtful and detailed comments. We have revised this manuscript carefully based on the comments. In particular, Fig. 2, Fig. 4, Fig. 7 and Fig 9 were redrawn to facilitate the readers with color vision deficiencies as suggested by the editorial team. Fig.4F1-F5 and Fig. 9F1-F5 were adjusted to better match the description in the captions and Fig. S6 was adjusted by using log scales as suggested by the reviewer.

Below we respond to the individual comments.

Reviewer #1

**Question**: The comments of the reviewers have been addressed, and the paper has been improved so that it is clearer and more readable. The analysis is substantially more thorough than in the original manuscript, and the results are now more robust and more useful. However, there are still some weaknesses that need to be addressed before the paper is appropriate for publication.

**Answer**: Thank the reviewer again for the constructive comments! The manuscript has been revised based on the comments.

**Question**: Line 57: "combination of multiple species": please specify the species included in Ox, to help readers who are not familiar with GEOS-Chem. (This information is provided on line 200, but would be better at this earlier point).

**Answer**: The definition of $O_x$ was moved forward to the Introduction Section.

**Question**: Line 185: How are the archived PO3 and LO3 terms applied to the single tracer? Is the LO3 term kept as a first-order loss term (LO3/[O3]) so that the original [O3] does not need to be archived? A sentence or two on the methods is needed for publication in GMD.

**Answer**: As indicated by the reviewer, the $LO_3$ that was read in the single $O_3$ tracer simulation is $LO_3/[O_3]$. The method has been clarified in the revised version (Lines 171-173):

"The GEOS-Chem full chemistry simulations with the updated KPP module were then performed to produce $PO_3$ (unit kg cm$^{-3}$ s$^{-1}$) and relative $LO_3$ (i.e., $LO_3/[O_3]$ with unit cm$^{-3}$ s$^{-1}$) every 20 minutes".

**Question**: Tagged-Ox exceeds Tagged-O3, but this is because the additional NOx/NOy species are included. The concentrations will be a lot closer if these are removed! The comparison doesn't indicate that the Tagged-O3 run is better, merely that it is more suitable for direct comparison with observed O3.

**Answer**: The major difference between the tagged-$O_3$ and tagged-Ox is emphasized in the revised version:

"it may not be an ideal choice to perform $O_3$ simulations based on the tagged-$O_x$ mode because $O_x$ is the combination of multiple species ($O_x=O_3+NO_2+2NO_3+3N_2O_5+HNO_3+HNO_4+$peroxyacylnitrates) and thus cannot be accurately compared with $O_3$ observations" in the Introduction Section.

"In contrast, the $O_x$ concentrations provided by the tagged-$O_x$ mode are higher than the $O_3$ concentrations by approximately 6 ppb, and the relative difference can reach 40% in the winter, which is thus not suitable for direct comparison with observed $O_3$" in the Conclusion Section.

**Question**: It is not surprising that the Tagged-O3 run can match the full model run, as PO3 and LO3 are derived directly from this run. A better test of the tagged mode would need to explore what happens under ozone changes (e.g., those associated with assimilation), as with increasing ozone changes the approach will lose accuracy. The approach described is good for representing near-current conditions, but is not suitable under substantially different conditions; the paper should be clear about this limitation.

**Answer**: Thank the reviewer for pointing out this issue! The limitation of the tagged-$O_3$ simulation was elaborated in the Conclusion Section: "Despite these advantages, it should be noted that the linear chemistry assumption by reading the archived $PO_3$ and $LO_3$ implies single $O_3$ tracer mode is good for representing near-current $O_3$ chemical conditions, particularly, for scientific issues associated with the sources and transport of tropospheric $O_3$ as well as assimilations in this work and the companion paper (Zhu et al., 2023). More cautious applications are suggested under substantially different $O_3$ chemical conditions as the linear chemistry assumption could not be satisfied".

**Question**: Fig 4 caption: note that F1-F5 show the effect of *removing* O3 formation over the NCP; the "effects of O3 formation" would be given by showing #1-#4. Note that a less extreme color scale (e.g., 40 to -40) would show up the effects outside the NCP more clearly.

**Answer**: Fig. 4F1-F5 and Fig. 9F1-F5 have been redrawn by showing experiments #1-#4. The color scale was adjusted to 40 to -40.

**Question**: Fig 5: 5 regions are defined in Fig 1, but 6 regions are shown here; it would be helpful to remind the reader that "E. China" refers to the whole domain of interest here.

**Answer**: Thank the reviewer for this suggestion! The domain definition of E. China was reminded in the captions of Table 2 and Fig. 5 in the revised version.

**Question**: The final section is a little short on analysis; results are described, but the consequences and implications of them are not identified or explored in much detail.

**Answer**: The discussion in the Conclusion Section has been adjusted to emphasize more on the consequences and implications of the analysis.

**Question**: Note that interannual variability should not affect long-term trends, it just affects assessment of trends over very short time periods such as the 6 years considered here.

**Answer**: We agree with the reviewer that it could be more accurate to use the term "interannual trends in background $O_3$". However, the interannual trend of background $O_3$ is not the target of this work and is not evaluated in our analysis. Consequently, we use the term "interannual variability in background $O_3$" because we found it may not be robust enough to make a conclusion about the interannual trends of background $O_3$.

Supplement

**Question**: Uncertainty analysis: "...randomly drawing N data points from the full set of N data points...". Should this be "with replacement"? If so, there is duplication. If not, the sampling just uses all points. Some clarification on the method used is needed here.

**Answer**: The bootstrapping method allows drawing individual data points multiple times to represent the error due to random sampling. It has been clarified in the revised description: "individual data points may be drawn multiple times".

**Question**: Fig S6: given the mass units of PO3, it would be better to present these graphs on a log-log scale (i.e., use a log scale on the X-axis, too).

**Answer**: Thank the reviewer for this suggestion! Fig. S6 has been adjusted by using log scales.

---

## Author Response (AR3)

We thank the editorial team for the valuable suggestions! Below we respond to the individual suggestions.

**Question**: PAN stands for peroxyacetyl nitrate (you misspelled it in line 57 of the track-changes manuscript). Also does this include other nitrates (methyl nitrate, PPAN) that play similar roles to PAN?

**Answer**: The Ox family in the GEOS-Chem model is defined as:

$Ox = O_3 + NO_2 + 2NO_3 + PAN + PPN + HNO_4 + 3N_2O_5 + HNO_3 + BrO + HOBr + BrNO_2 + 2BrNO_3 + MPN + ETHLN + MVKN + MCRHN + MCRHNB + PROPNN + R4N2 + PRN1 + PRPN + R4N1 + HONIT + MONITS + MONITU + OLND + OLNN + IHN1 + IHN2 + IHN3 + IHN4 + INPB + INPD + ICN + 2IDN + ITCN + ITHN + ISOPNOO1 + ISOPNOO2 + INO2B + INO2D + INA + IDHNBOO + IDHNDOO1 + IDHNDOO2 + IHPNBOO + IHPNDOO + ICNOO + 2IDNOO + MACRNO2 + ClO + HOCl + ClNO_2 + 2ClNO_3 + 2Cl_2O_2 + 2OClO + IO + HOI + IONO + 2IONO_2 + 2OIO + 2I_2O_2 + 3I_2O_3 + 4I_2O_4$

As indicated in this question, the Ox family includes other species similar to PAN. The usage of the term "peroxyacylnitrates" follows previous literature (e.g., Zhang et al., 2008, Zhu et al., 2017 and Han et al., 2018). It has been changed to "PANs" in the revised version.

References:

Zhang, L., Jacob, D. J., Boersma, K. F., Jaffe, D. A., Olson, J. R., Bowman, K. W., Worden, J. R., Thompson, A. M., Avery, M. A., Cohen, R. C., Dibb, J. E., Flock, F. M., Fuelberg, H. E., Huey, L. G., McMillan, W. W., Singh, H. B., and Weinheimer, A. J.: Transpacific transport of ozone pollution and the effect of recent Asian emission increases on air quality in North America: an integrated analysis using satellite, aircraft, ozonesonde, and surface observations, Atmos Chem Phys, 8, 6117-6136, DOI 10.5194/acp-8-6117-2008, 2008.

Zhu, Y., Liu, J., Wang, T., Zhuang, B., Han, H., Wang, H., Chang, Y., and Ding, K.: The Impacts of Meteorology on the Seasonal and Interannual Variabilities of Ozone Transport From North America to East Asia, J Geophys Res-Atmos, 122, 10,612-610,636, 10.1002/2017jd026761, 2017.

Han, H., Liu, J., Yuan, H., Zhuang, B., Zhu, Y., Wu, Y., Yan, Y., and Ding, A.: Characteristics of intercontinental transport of tropospheric ozone from Africa to Asia, Atmos Chem Phys, 18, 4251-4276, 10.5194/acp-18-4251-2018, 2018.

**Question**: Please ensure that the colour schemes used in your maps and charts allow readers with colour vision deficiencies to correctly interpret your findings. Please check your figures using the Coblis – Color Blindness Simulator (https://www.color-blindness.com/coblis-color-blindness-simulator/) and revise the colour schemes accordingly.

**Answer**: Fig. 2, Fig. 4, Fig. 7 and Fig 9 were redrawn by using color tables "BlueWhiteOrangeRed" and "NCV_blu_red". Maps and charts have been rechecked to ensure they are using simple blue-red color schemes to facilitate the readers with color vision deficiencies.